# Quantifying vegetation indices using Terrestrial Laser Scanning: methodological complexities and ecological insights from a Mediterranean forest

William Rupert Moore Flynn[1], Harry Jon Foord Owen[2], Stuart William David Grieve[1,3] and Emily Rebecca Lines[2]

[1]School of Geography, Queen Mary University of London, Mile End Rd, Bethnal Green, London E1 4NS

[2]Department of Geography, University of Cambridge, Downing Place, Cambridge, CB2 3EN

[3]Digital Environment Research Institute, Queen Mary University of London, New Road, London, E1 1HH

*Correspondence to: W. R. M. Flynn (w.r.m.flynn@qmul.ac.uk)*

**Abstract.** Accurate measurement of vegetation density metrics including plant, wood and leaf area indices (PAI, WAI and LAI) is key to monitoring and modelling carbon storage and uptake in forests. Traditional passive sensor approaches, such as Digital Hemispherical Photography (DHP), cannot separate leaf and wood material, nor individual trees, and require many assumptions in processing. Terrestrial Laser Scanning (TLS) data offer new opportunities to improve understanding of tree and canopy structure. Multiple methods have been developed to derive PAI and LAI from TLS data, but there is little consensus on the best approach, nor are methods benchmarked as standard.

Using TLS data collected in 33 plots containing 2472 trees of five species in Mediterranean forests, we compare three TLS methods (*LiDAR Pulse*, *2D Intensity Image* and *Voxel-Based*) to derive PAI and compare with co-located DHP. We then separate leaf and wood in individual tree point clouds to calculate the ratio of wood to total plant area (α), a metric to correct for non-photosynthetic material in LAI estimates. We use individual tree TLS point clouds to estimate how α varies with species, tree height and stand density.

We find the *LiDAR Pulse* method agrees most closely with DHP, but is limited to single scan data so cannot determine individual tree properties, including α. The *Voxel-Based* method shows promise for ecological studies as it can be applied to individual tree point clouds. Using the *Voxel-Based* method, we show that species explain some variation in α, however, height and plot density were better predictors.

Our findings highlight the value of TLS data to improve fundamental understanding of tree form and function, but also the importance of rigorous testing of TLS data processing methods at a time when new approaches are being rapidly developed. New algorithms need to be compared against traditional methods, and existing algorithms, using common reference data. Whilst promising, our results show that metrics derived from TLS data are not yet reliably calibrated and validated to the extent they are ready to replace traditional approaches for large scale monitoring of PAI and LAI.

## 1 Introduction

Leaf Area Index (LAI), defined as half the amount of green leaf area per unit ground area (Chen and Black, 1992), determines global evapotranspiration, phenological patterns and canopy photosynthesis, and is therefore an essential climate variable (ECV), as well as a key input in dynamic global vegetation models (Sea et al., 2011; Weiss et al., 2004). Accurate measurements of leaf, wood and plant area indices (LAI, WAI and PAI) have historically been derived from labour intensive destructive sampling (Baret et al., 2013; Jonckheere et al., 2004), so over large spatial or temporal scales these can only be measured indirectly, typically with remote sensing. Large-scale remote sensing, using spaceborne and airborne instruments, has been widely used to estimate LAI over large areas (Pfeifer et al., 2012), but requires calibration and validation using in situ measurements to constrain information retrieval (Calders et al., 2018). Non-destructive in situ vegetation index estimates have historically been made by measuring light transmission below the canopy and using simplifying assumptions about canopy structure to estimate the amount of intercepting material (e.g. Beer-Lambert's law; Monsi and Saeki, 1953). The most common method, Digital Hemispherical Photography (DHP; Figure 1a), requires both model assumptions and subjective user choices during data acquisition and processing in order to estimate both PAI and LAI (Breda, 2003). DHP images are processed by separating sky from canopy, but not photosynthetic from non-photosynthetic vegetative material, so additional assumptions are needed to calculate either LAI or WAI (Jonckheere et al., 2004; Pfeifer et al., 2012). Separation of LAI from PAI can be achieved by removing or masking branches and stems from hemispherical images (e.g. Sea et al., 2011; Woodgate et al., 2016), but is not reliable when leaves are occluded by woody components (Hardwick et al., 2015). An alternative approach is to take separate DHP measurements in both leaf on and leaf off conditions, and derive empirical wood to plant ratios (WAI/PAI, $\alpha$) (Leblanc and Chen, 2001), but this is not always practical, for example in evergreen forests. The difficulty of separation means that studies often omit correcting for the effect of WAI on optical PAI measurements altogether (Woodgate et al., 2016), but since woody components in the forest canopy can account for more than 30% of PAI (Ma et al., 2016) this can introduce overestimation. Further, although DHP estimates of LAI or PAI are valuable both for ecosystem monitoring and developing satellite LAI products (Hardwick et al., 2015; Pfeifer et al., 2012), they are limited to sampling only at a neighbourhood or plot level (Weiss et al., 2004), and cannot be used to measure individual tree LAI except for open grown trees (Béland et al., 2014).

The ratio of wood to total plant area, $\alpha$, is known to be dynamic, changing in response to abiotic and biotic conditions. For example, the Huber value (sapwood to leaf area ratio, a related measure to $\alpha$) may vary according to water availability (Carter and White, 2009). Leaf area may therefore be indicative of the drought tolerance level of a tree, with more drought tolerant species displaying a lower leaf area, reducing the hydraulic conductance of the whole tree and therefore increasing its drought tolerance (Niinemets and Valladares, 2006). $\alpha$ has been hypothesised to increase with the size of a tree in response to the increased hydraulic demand associated with greater hydraulic resistance of tall trees (Magnani et al., 2000) and higher transpiration rates of larger LAI (Battaglia et al., 1998; Phillips et al., 2003). Stand density may also impact $\alpha$ (Long and Smith, 1988; Whitehead, 1978), as increased stand level water use scales linearly with LAI (Battaglia et al., 1998; Specht and Specht, 1989), reducing water availability to individual trees competing for the same resources (Jump et al., 2017). Large scale quantification of $\alpha$ or Huber value, however, is difficult as studies usually rely on a small number of destructively sampled trees (e.g. Carter and White, 2009; Magnani et al., 2000), litterfall traps (e.g. Phillips et al., 2003) or

masking hemispherical images (e.g. Sea et al., 2011; Woodgate et al., 2016). These approaches are only applicable
on a small to medium scale, and in the case of image masking, cannot differentiate between individuals. Variation
in $\alpha$, for example by species and or stand structure, is therefore largely unknown.

**1.1 TLS methods for calculating PAI, LAI and WAI**

Terrestrial Laser Scanning (TLS) generates high-resolution 3D measurements of whole forests and individual
trees (Burt et al., 2018; Disney, 2018), leading to the development of completely new monitoring approaches to
understand the structure and function of ecosystems (Lines et al., 2022). Unlike traditional passive sensors, TLS
can estimate PAI, WAI, and LAI for both whole plots and individual tree point clouds (Calders et al., 2018), and
is unaffected by illumination conditions. This has led to the development of several methods for processing TLS
data to extract the key metrics PAI, WAI and LAI (e.g. Hosoi and Omasa, 2006; Jupp et al., 2008; Zheng et al.,
2013). However, intercomparison studies of algorithms and processing approaches to derive the same metrics
from different TLS methods are lacking.TLS methods for extracting PAI, LAI and WAI can be broadly
categorised into two types: (1) LiDAR return counting, using single scan data (e.g., the *LiDAR Pulse* method;
Jupp et al., 2008, and *2D Intensity Image* method; Zheng et al., 2013) and (2) point cloud voxelisation, usually
using co-registered scans (e.g., the *Voxel-Based* method; Hosoi and Omasa, 2006).
The *LiDAR Pulse* method (Jupp et al., 2008; Figure 1b) estimates gap fraction ($P_{gap}$) using single scan data, as a
function of the total number of outgoing LiDAR pulses from the sensor and the number of pulses that are
intercepted by the canopy. This method, which eliminates illumination impacts associated with the use of DHP
(Calders et al., 2014), has been implemented in the python module, *PyLidar* ([www.pylidar.org](www.pylidar.org)) and the R package,
*rTLS* (Guzman, et al. 2021). Using the *LiDAR Pulse* method, Calders et al. (2018) compared PAI estimates from
two ground-based passive sensors (LiCOR LAI-2000 and DHP) with TLS data collected with a RIEGL VZ-400
TLS in a deciduous woodland, and found the two passive sensors underestimated PAI values compared to TLS,
with differences dependent on DHP processing and leaf on/off conditions.
The *2D Intensity Image* method (Zheng et al., 2013; Figure 1c), also uses raw single scan TLS point clouds, but,
unlike the *LiDAR Pulse* method, converts LiDAR returns into 2D panoramas where pixel values represent return
intensity. PAI is estimated by classifying pixels as sky or vegetation, based on their intensity value, to estimate
$P_{gap}$, and then applying Beer-Lambert's law. Like the *LiDAR Pulse* method, this approach has been shown to
generate higher PAI estimates than DHP (Calders et al., 2018; Woodgate et al., 2015; Grotti et al., 2020), with
differences attributed to the greater pixel resolution and viewing distance of TLS resolving more small canopy
details (Grotti et al., 2020).
The *Voxel-Based* method (Figure 1d) estimates PAI by segmenting a point cloud into voxels and either simulating
radiative transfer within each cube (Béland et al., 2014; Kamoske et al., 2019), or classifying voxels as either
containing vegetation or not, and dividing vegetation voxels by the total number of voxels (Hosoi and Omasa,
2006; Itakura and Hosoi, 2019; Li et al., 2017). Crucially, this method may be applied to multiple co-registered
scan point clouds and so can be used to calculate PAI for both whole plots and individual, segmented TLS trees.
However, PAI estimates derived using the voxel method are highly dependent on voxel size (Calders et al., 2020).
Using a radiative transfer approach, Béland et al. (2014) demonstrated that voxel size is dependent on canopy
clumping, radiative transfer model assumptions and occlusion effects, making a single, fixed choice of voxel size
for all ecosystem types, scanners or datasets impossible. To test various approaches to selecting voxel size using
a voxel classification approach, Li et al. (2016) matched voxel size to point cloud resolution, individual tree leaf
size, and minimum beam distance and tested against destructive samples, finding that voxel size matched to point
cloud resolution had the closest PAI values to destructive samples. The *LiDAR Pulse* method and *2D Intensity*
*Image* method both use single scan data. However, to generate robust estimates of canopy properties that avoid
errors from occlusion effects, multiple co-registered scans taken from different locations are likely needed (Wilkes
et al., 2017). Further, both these methods require raw unfiltered data to accurately measure the ratio of pulses
emitted from the scanner and number of pulses that are intercepted by vegetation. This means "noisy" points

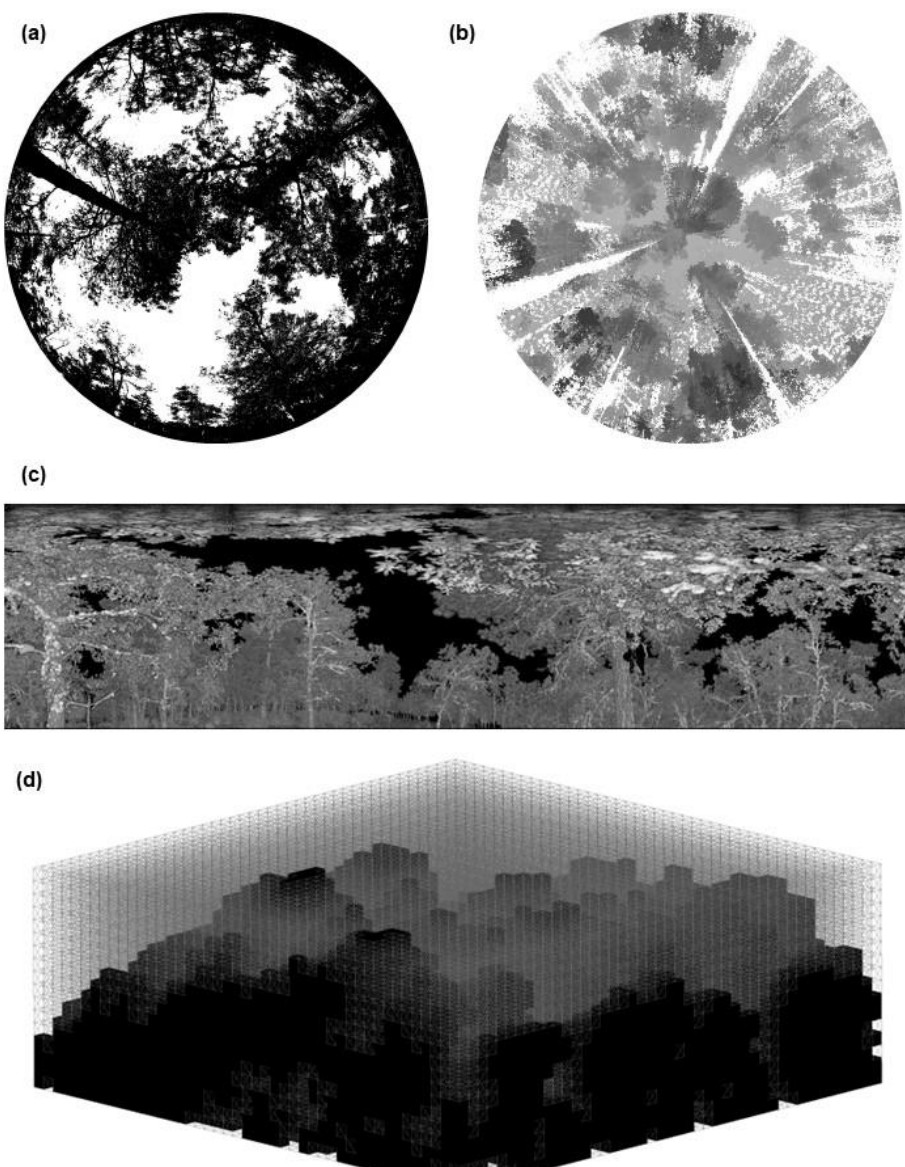

caused by backscattered pulses (Wilkes et al., 2017) are included in analyses, potentially leading to higher PAI
estimates. However, the *LiDAR Pulse* and *2D Intensity Image* methods may introduce fewer estimation errors
compared to DHP, which is influenced by differences in sky illumination conditions and camera exposure (Weiss
et al., 2004).


**Figure 1: Visual representation of the four methods for PAI and WAI estimation used in this study: (a) a binarised digital hemispherical photograph (DHP), (b) TLS raw single scan point cloud, for the *LiDAR Pulse* method (Jupp et al., 2008). Image shows a top-down view of raw point cloud and greyscale represents low (grey) and high (black) Z values, (c) TLS 2D intensity image for the *2D Intensity Image* method (Zheng et al., 2013), (d) Voxelised co-registered whole plot point cloud for the *Voxel-Based* method (Hosoi and Omasa, 2006), showing a representative schematic of cube voxels with edge length of 1m, voxelised using the *R package VoxR* (Lecigne et al., 2018). Solid black voxels are classified as containing vegetation (filled) and voxels outlined with grey lines are voxels classified as empty.**

**1.2 Scope and aims**

The aims of this study are twofold: the first aim is to compare three TLS methods for estimating PAI with traditional DHP. The second aim of this study is to use TLS to understand drivers of individual tree $\alpha$ variation.

In this study we use a dataset of 528 co-located DHP and high-resolution TLS scans from 33 forest plots to compare DHP derived PAI ($PAI_{DHP}$) with estimates from three methods to estimate PAI from TLS data ($PAI_{TLS}$): the *LiDAR Pulse* method; the *2D Intensity Image* method and the *Voxel-Based* method (Figure 1). We use a dataset collected from a network of pine/oak forest plots in Spain (Owen et al., 2021) and ask **(1)** are the three TLS methods able to reproduce $PAI_{DHP}$ estimates at single scan and whole plot level? **(2)** does $\alpha$, calculated from the *Voxel-Based* method on individual tree point clouds, vary with species and tolerance to drought? and **(3)** does $\alpha$ scale with height and stand density?

**2. Methods**

**2.1 Study site**

We collected TLS and DHP data from 29 plots in Alto Tajo Natural Park (40°41′N, 02°03′W; FunDIV – Functional Diversity plots; see Baeten et al. (2013) for a detailed description of the plots) and four plots in Cuellar (41°23′N 4°21′W) in June - July 2018 (see Owen et al. (2021) for full details) (Figure A1). Plots contained two oak species: semi-deciduous *Q. faginea* and evergreen *Q. ilex,* and three pine species: *P. nigra, P. pinaster* and *P. sylvestris*. *P. sylvestris* is the least drought tolerant species, followed by *P. nigra, Q. faginea*, *Q. ilex*; shade tolerance follows the same ranking (Niinemets and Valladares, 2006; Owen et al., 2021). Although not quantitatively ranked, *P. pinaster* has been shown to be very drought tolerant, appearing in drier areas than the other species (Madrigal-González et al., 2017). The area is characterised by a Mediterranean climate (altitudinal range 840 – 1400 m.a.s.l.) (Jucker et al., 2014; Madrigal-González et al., 2017). In addition to the five main canopy tree species, plots contained an understory of *Juniperus thurifera* and *Buxus sempervirens* (Kuusk et al., 2018).

**2.2 Field protocol**

In each of the 33 plots of size 30 x 30 m, we collected TLS scans on a 10 m grid, making 16 scan locations following Wilkes et al. (2017) to minimise occlusion effects associated with insufficient scans. We used a Leica HDS6200 TLS set to super high resolution (3.1 x 3.1mm resolution at 10 m with a beam divergence of $\leq$ 5 mm at 50 m; scan time 6m 44 s; see Owen et al. (2021)). At each of the 528 scan locations and following the protocol in Pfeifer et al. (2012), we captured co-located DHP images with three exposure settings (automatic and $\pm$ one stop exposure compensation), levelling a Canon EOS 6D full frame DSLR sensor with a Sigma EX DG F3.5 fisheye lens, mounted on a Vanguard Alta Pro 263AT tripod.

## 2.3 Calculation of single scan and whole plot PAI using DHP data

For each of the red-green-blue (RGB) DHP images we extracted the blue band for image thresholding, as this best represents sky/vegetation contrast (Pfeifer et al., 2012). For each plot, we picked the exposure setting that best represented sky/ vegetation difference based on pixel brightness histograms of four sample locations indicative of the plot. We carried out automatic image thresholding using the Ridler and Calvard method (1978), to create a binary image of sky and vegetation, avoiding subjective user pixel classification (Jonckheere et al., 2005). We calculated PAI from the binary image, limiting the field of view to a 5° band centred on the hinge angle of 57.5° (55° – 60°). The hinge angle has a path length through the canopy twice the canopy height, so the band around it is an area of significant spatial averaging taken as representative of canopy structure of the area (Calders et al., 2018; Jupp et al., 2008). From the binarised hinge angle band we calculated $P_{gap}$ as the number of sky pixels divided by the total number of pixels and PAI using an inverse Beer-Lambert law equation (Monsi and Saeki, 1953). We calculated whole plot PAI as the arithmetic mean of the 16 plot scan location PAI estimates. As this value does not correct for canopy clumping, it is better described as effective PAI, rather than true PAI (Woodgate et al., 2015). However, as the TLS and DHP methods we apply here account for canopy clumping differently, we compared effective values and here-on refer to effective PAI as PAI (Calders et al., 2018). DHP images used in this study are freely available (see Flynn et al., 2023).

## 2.4 Calculation of single scan and whole plot PAI from TLS data

To calculate PAI using the *LiDAR Pulse* method (Jupp et al., 2008), we calculated $P_{gap}$ for a single scan (Figure 1b) by summing all returned laser pulses and dividing by the number of total outgoing pulses, following Lovell et al. (2011; see Eq. 7 in that study), and then estimated PAI following Jupp et al. (2008; see Eq. 18 in that study), setting the sensor range to 5° around the hinge angle as before (55° – 60°). Single scan PAI was taken as the cumulative sum of PAI values estimated by vertically dividing the hinge region into 0.25 m intervals (Calders et al., 2014). We implemented the *LiDAR Pulse* method using the open-source *R* (R Core Team, 2020) package, *rTLS* (Guzmán and Hernandez, 2021).

To calculate PAI using the *2D Intensity Image* method (Zheng et al., 2013)*,* we converted 3D TLS point cloud data from all 528 scan locations into polar coordinates, scaled intensity values to cover the full 0-255 range (Figure 1c) and rasterised into a 2D intensity image using the open-source *R* package, *raster* (Hijmans, 2022). We cut the 2D intensity image to a 5° band around the hinge angle (55° – 60°) and classified sky and vegetation pixels in each image using the Ridler and Calvard method (1978). We calculated $P_{gap}$ as the number of pixels classified as sky divided by the total number of pixels and derived PAI with an inverse Beer-Lambert law equation (Monsi and Saeki, 1953).

Following the same approach as applied to our DHP data, we calculated whole plot PAI for the *LiDAR Pulse* and *2D Intensity Image* methods as the arithmetic mean of the 16 plot scan location PAI estimates.

To calculate PAI using the *Voxel-Based* method, we followed a voxel classification approach (Hosoi and Omasa, 2006), downsampling the point cloud to 0.05 m to aid computation time and matching the voxel size to the resolution of the point cloud, following Li et al. (2016), who showed that matching the voxel size to the point cloud point to point minimum distance (resolution) increases accuracy as small canopy gaps are not included in voxels classified as vegetation. We chose to use a voxel classification approach (rather than a radiative transfer

based one) as this method is widely applicable to a range of TLS systems and levels of processing, as well as providing explicit guidance on voxel size selection, which is known to impact derived PAI estimates (Li et al., 2016). We re-combined individually segmented trees, filtered for noise using a height-dependent statistical filter (see Owen et al., 2021) back into whole plot point clouds and voxelised them using the open source *R* package, *VoxR* (Lecigne et al., 2018), with a full grid covering the minimum to maximum XYZ ranges of the plot. We classified any voxel containing > 0 points as vegetation ("filled"), and empty voxels as gaps. We then split the voxelised point cloud vertically into slices one voxel high. Within each slice, the contact frequency is calculated as the fraction of filled to total number of voxels. We then multiplied the contact frequency by a correction factor for leaf inclination, set at 1.1 (Li et al., 2017), and whole plot PAI was calculated as the sum of all slices' contact frequencies.

**2.5 Calculation of individual tree PAI, WAI and α using the voxel-based method**

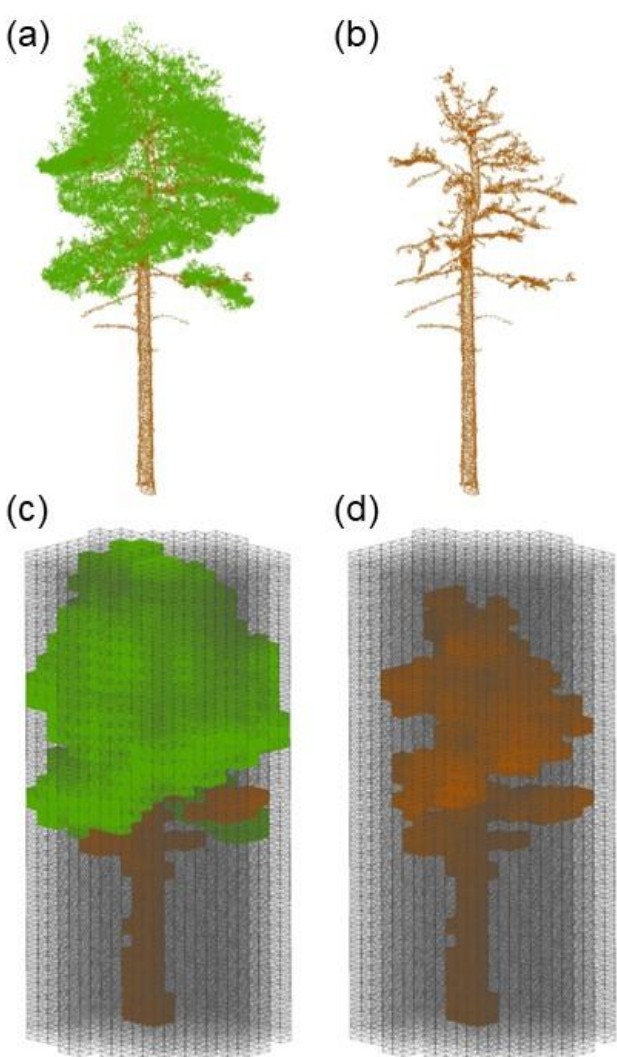

**Figure 2: Visualisation of the workflow for applying the Voxel-Based method to estimate individual-tree PAI, WAI and α. (a) Individual tree point cloud; (b) separated leaf off (wood) individual tree point cloud; (c) voxelised individual tree point cloud; (d) voxelised wood cloud. Coloured voxels (green represents leaf and brown represents wood) are filled voxels and grey lines are empty voxels. Empty voxels occupy the space within the projected crown area of the tree. Image shows schematic of point cloud voxelised with cube voxels with edge length of 0.5 m. Panels (a) and (b) show wood and leaf separation of an example *P. sylvestris*, carried out using *TLSeparation* (Vicari et al., 2019). Point cloud**

**216** **voxelisation was carried out using modified functions from *R* package *VoxR* (Lecigne et al., 2018). Note that our method**
**217** **used voxel sizes at the resolution of the cloud (0.05 m), but here we present an image with larger voxels to ease visual**
**218** **interpretation.**

**219** As the only method using multiple co-registered scans, the *Voxel-Based* method is only method compared in this

**220** study capable of deriving PAI, WAI and LAI of segmented individual tree point clouds. We estimated PAI and

**221** WAI for 2472 individual trees segmented from co-registered point clouds following a similar method to the whole

**222** plot point cloud. We used individual tree point clouds downsampled to 0.05 m, to aid computation time, and

**223** segmented using the automated tree segmentation program *treeseg* (Burt et al., 2019), implemented in C++, by

**224** Owen et al. (2021) for that study. Individual segmented tree data used in this study are freely available (see Owen

**225** et al., 2022).

**226** To estimate PAI, WAI and α for each tree, we used individual tree point clouds wood – leaf separated by Owen

**227** et al. (2021) using the open source Python library *TLSeparation* (Vicari et al., 2019), and then used the separated

**228** wood point clouds to calculate WAI. *TLSeparation* assigns points as either leaf or wood, iteratively looking at a

**229** predetermined number of nearest neighbours (*knn*). The *knn* of each iteration is directly dependent on point cloud

**230** density, since high density point clouds will require higher a *knn* (Vicari et al., 2019). The utility package in

**231** *TLSeparation* was used to automatically detect the optimum *knn* for each tree point cloud.

**232** To voxelise individual tree complete (Figure 2a) and wood only (Figure 2b) point clouds, we used a modified

**233** approach based on Lecigne et al. (2018), voxelising within the projected crown area of the whole tree point cloud

**234** (Figure 2c) to calculate PAI. In the same way as for PAI, we calculated WAI using the separated wood point cloud

**235** within the projected crown area of the whole tree (Figure 2d; using the whole crown and not just the wood point

**236** cloud), and derived α for each tree as $WAI/_{PAI}$, allowing a comparison with existing literature estimating α for a

**237** range of ecosystems, (Sea et al., 2011; Woodgate et al., 2016).

**238** **2.6 Statistical Analyses**

**239** We tested the relationships between $PAI_{TLS}$ and $PAI_{DHP}$ estimates using Standardised Major Axis (SMA) using

**240** the open source *R* (R Core Team, 2020) package, *smatr* (Warton et al., 2012). SMA is an approach to estimating

**241** a line of best fit where we are not able to predict one variable from another (Warton et al., 2006); we chose SMA

**242** because we do not have a 'true' validation dataset, so avoid assuming either DHP or any of the TLS methods

**243** produces the most accurate results. For each TLS method, we assessed the relationship with DHP using the

**244** coefficient of determination and RMSE. We chose to compare PAI values rather than WAI or LAI as to do so

**245** would mean an additional correction for non-photosynthetic elements, which each method does in different ways,

**246** so introducing further source of uncertainty and limiting our ability to fairly compare processing approaches. To

**247** further understand observed drivers of variance in PAI, we tested the relationship between PAI and whole plot

**248** crown area index, CAI, a proxy measure of stand density and local competition (Caspersen et al., 2011; Coomes

**249** et al., 2012). We calculated CAI as the sum of TLS-derived projected crown area, divided by the plot area (Owen

**250** et al., 2021).

**251**

**252** To test if α differs by species, we used linear mixed models (LMMs) in the *R* package, *lme4* (Bates et al., 2015).

**253** We included an intercept only random plot effect to account for local effects on α:


$$\alpha_{i,sj} = \varphi_s + Plot_j, \qquad\qquad (1)$$

here, $\alpha_i$ is α of an individual of species $s$, in plot $j$, and $\varphi_s$ is the parameter to be fit. To test the effect of stand
structure and tree height on α, we fit relationships separately for each species, again including a random plot
effect:

$$\alpha_{i,sj} = \varphi_s + b_s\,H_i + c_s\,CAI_j + Plot_{sj}. \qquad\qquad (2)$$

here $H_i$ is the height of the tree, $CAI_j$ is the crown area index for the plot, with other parameters as before.
For each species' model (equation 2), we calculated the intra-class correlation coefficient (ICC). The ICC, similar
to coefficient of determination, quantifies the amount of variance explained by the random effect in a linear mixed
model (Nakagawa et al., 2017).
**3. Results**
**3.1 Comparison of plant area index estimated by DHP and single scan TLS**
Of the two single scan TLS methods tested (*LiDAR Pulse* method and *2D Intensity Image* method), we found that
the relationship between PAI estimated using the *LiDAR Pulse* method and $PAI_{DHP}$, had a higher $R^2$ than the *2D*
*Intensity Image* method (SMA; *LiDAR Pulse* method $R^2 = 0.50$, slope = 0.73, $p < 0.001$, RMSE = 0.14, and *2D*
*Intensity Image* method $R^2 = 0.22$, slope = 0.38, $p < 0.001$, RMSE = 0.39, respectively, Figure 3a). At larger PAI
values, both TLS methods underestimated PAI relative to DHP (Figure 3b). We found statistically significant
negative correlations between residuals and DHP for both methods (SMA; *2D Intensity Image* method residuals
$R^2 = 0.85$, slope = −0.88, $p < 0.01$; *LiDAR Pulse* method residuals $R^2 = 0.47$, slope = -0.70, $p < 0.01$; Figure 3b).
The *2D Intensity Image* method showed larger underestimation at higher $PAI_{DHP}$ values, suggesting this method
may saturate sooner for higher PAI values than either DHP or the *LiDAR Pulse* method (Figure 3b).

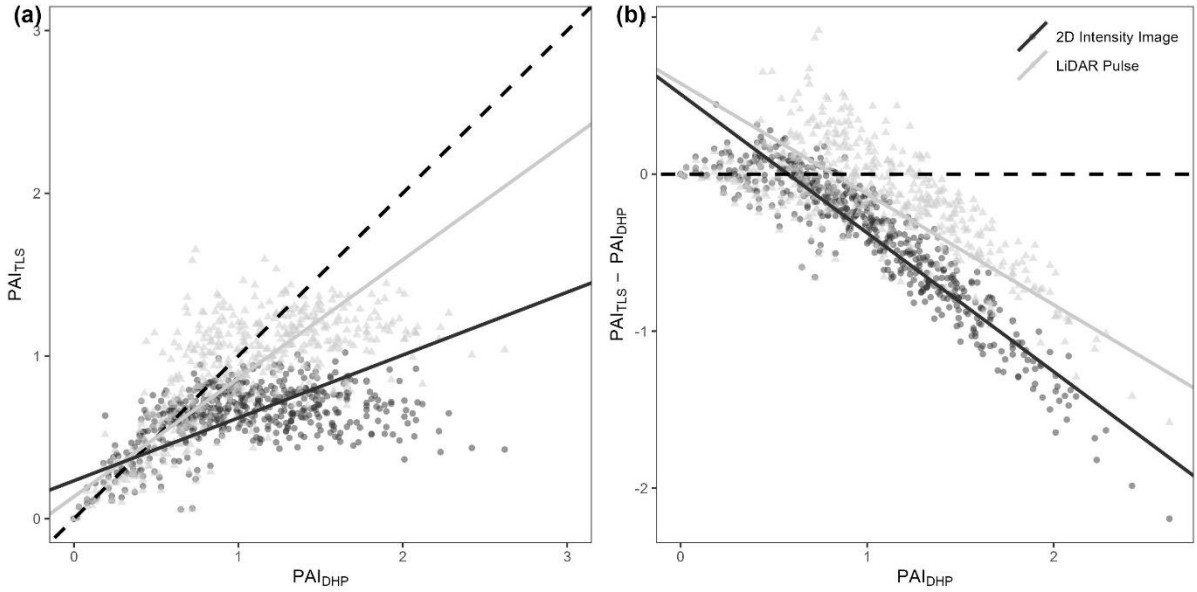


**Figure 3: Comparison of single scan PAI_TLS and PAI_DHP estimates, for all 528 scan locations (16 per plot). (a) The correlation between DHP derived PAI with PAI derived using the 2D Intensity Image method R² = 0.22, slope = 0.38, p < 0.001, RMSE = 0.39 (circles), and LiDAR Pulse method R² = 0.50, slope = 0.73, p < 0.001, RMSE = 0.14 (triangles). Dashed line in panel (a) represents 1:1 relationship. (b) The difference between PAI_TLS and PAI_DHP estimates for the 2D Intensity Image method, and LiDAR Pulse method. Dashed line in panel (b) represents 0. Solid lines show statistically significant relationships fitted using SMA (p < 0.01).**

## 3.2 Comparison of whole plot plant area index estimated using TLS and DHP and the effect of plot structure on PAI

We found statistically significant correlations between whole plot PAI_TLS values and PAI_DHP for all three TLS methods (Figure 4). As for single scans, the *LiDAR Pulse* method showed the closest agreement to PAI_DHP, here compared to both the *Voxel-Based* and *2D Intensity Image* methods (SMA; *LiDAR Pulse* method R² = 0.66, slope = 0.82, p < 0.01, RMSE = 0.14; *Voxel-Based* method R² = 0.39, slope = 2.76, p < 0.01, RMSE = 0.88; *2D Intensity Image* method R² = 0.35, slope = 0.36, p < 0.01, RMSE = 0.39, respectively; Figure 4a). The *2D Intensity Image* method and *LiDAR Pulse* method consistently underestimated PAI compared to DHP, whilst the *Voxel-Based* method underestimated in plots with lower PAI_DHP and overestimated in plots with higher PAI_DHP. The *Voxel-Based* method's high PAI values compared to other methods is likely due to its use of multiple co-registered scans reducing occlusion effects prevalent in single scan data.

To assess the effect of plot structure on variation in TLS derived PAI, we compared PAI_TLS estimates CAI (Figure 4b). We found a significant positive relationship between CAI and PAI estimated using each of the *LiDAR Pulse* method, the *Voxel-Based* method, and DHP (SMA*; LiDAR Pulse* method R² = 0.79, slope = 1.69, p < 0.01; *Voxel-Based* method R² = 0.76, slope = 5.72, p < 0.01; *2D Intensity Image* method R2 = 0.15, slope = 0.76, p < 0.05; DHP R² = 0.46, slope = 2.07, p < 0.01, respectively; Figure 4b), where the *2D Intensity Image* method shows signs of saturation at medium CAI values (Figure 4b).

302

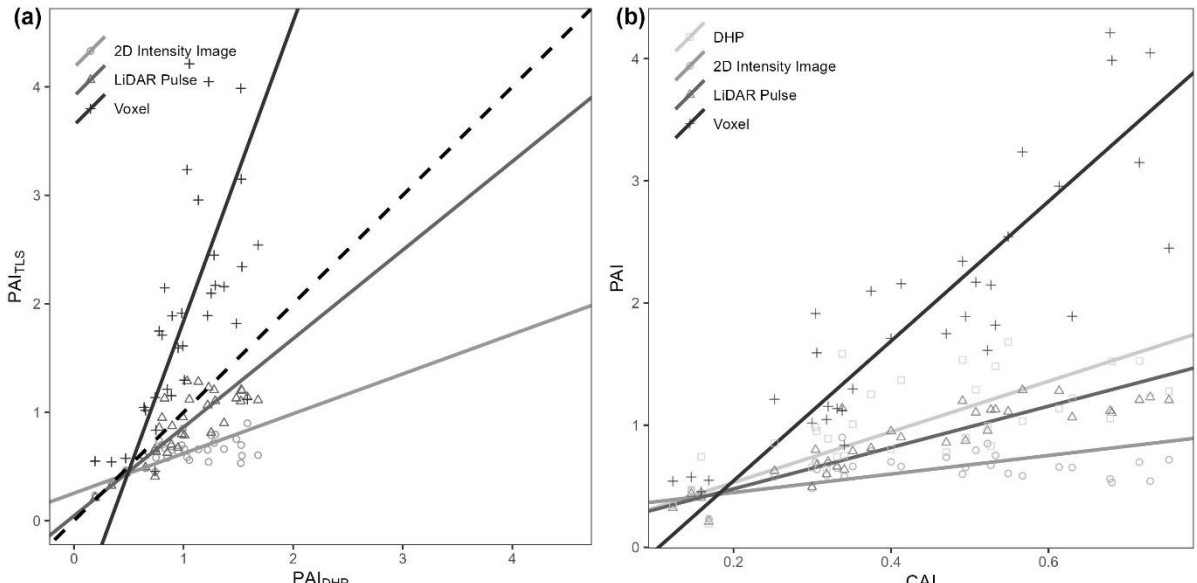

**Figure 4: Comparison of plot level PAI$_{TLS}$ vs PAI$_{DHP}$, and CAI vs PAI estimates for all 33 plots. (a) The correlation between DHP derived PAI and PAI derived using 2D Intensity Image R$^2$ = 0.35, slope = 0.36, p < 0.01, RMSE = 0.39 (circle), LiDAR Pulse R$^2$ = 0.66, slope = 0.82, p < 0.01, RMSE = 0.14 (triangle) and Voxel-Based R$^2$ = 0.39, slope = 2.76, p < 0.01, RMSE = 0.88 (cross) methods (b) The correlation between TLS derived CAI and PAI derived using DHP R$^2$ = 0.46, slope = 2.07, p < 0.01 (square), 2D Intensity Image R$^2$ = 0.15, slope = 0.76, p < 0.05 (circle) LiDAR Pulse R$^2$ = 0.79, slope = 1.69, p < 0.01 (triangle) and Voxel-Based R$^2$ = 0.76, slope = 5.72, p < 0.01 (cross) methods. Lines show statistically significant relationships fitted using SMA (p < 0.01). Dashed line in panel (a) represents 1:1 relationship.**

### 3.4 Influence of species, tree height and CAI on α

To understand drivers of variance in α, we used individual tree PAI and WAI, calculated using the *Voxel-Based* method to test the relationship between species and α, and height/ CAI and α. We found that more drought tolerant species generally had higher α values than less drought tolerant species (Table B1; Figure 5), however, confidence intervals were wide and overlapping, suggesting that species is not a strong predictor of variation in α. We found a statistically significant negative effect of height (p < 0.001; Table B2; Figure 6a) and positive effect of CAI (p < 0.01 – 0.05; Table B2; Figure 6b) on α for all species apart from *P. sylvestris*. α decreased more rapidly with height and increased less rapidly with CAI for oaks than pines. Statistically significant ICC values were higher for *P. nigra* (ICC = 0.211; Table B2) than *P. pinaster, Q. faginea* and *Q. ilex* (ICC = 0.036; 0.060; 0.070, respectively), showing that more α variation is explained by the random plot effect in *P. nigra* than the other species. *P. pinaster* has a wider confidence interval (Figure 5), possibly explained by its lower sample size. To understand drivers of variance in WAI we carried out additional analysis to test the relationship between WAI and species, height, CAI, and PAI, and presented these results in Appendix C (Figure C3; Tables C3, C4).

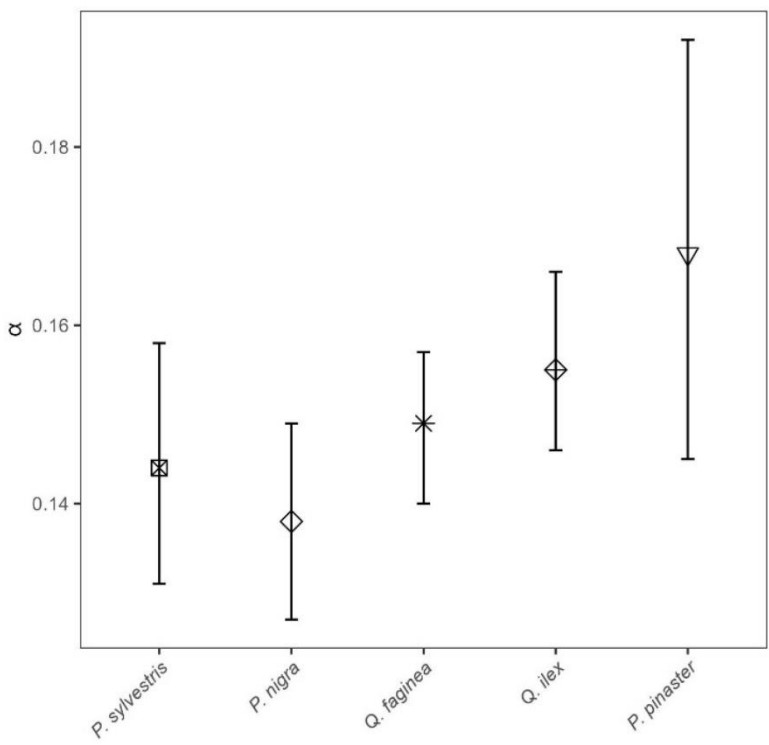

**Figure 5: Linear mixed model derived α values (φ, equation 1) for all 2472 individual trees of species *P. sylvestris, P. nigra, Q. faginea, Q. ilex* and *P. pinaster*. Error bars represent 95% confidence intervals. Species are listed left to right from low – high drought tolerance, with the exception of *P. pinaster*, for which drought tolerance index has not been calculated in the literature. Drought tolerance rankings are taken from Niinemets and Valladares (2006).**

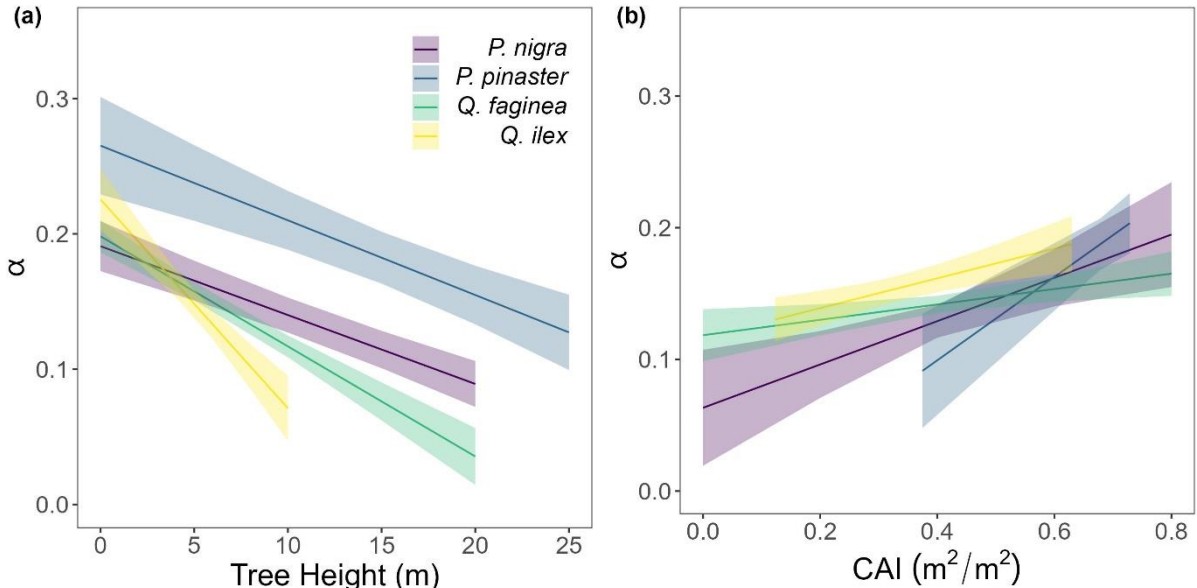

**Figure 6: Variation in α for each species: *Pinus nigra, P. pinaster, Q. faginea* and *Q. ilex* with (a) height and (b) plot CAI. Lines represent statistically significant linear mixed models (equation 2; significance levels from p < 0.001 to p < 0.05). Ribbons represent 95% confidence intervals. The model for *P. sylvestris* was not statistically significant.**

**4. Discussion**

**4.1 Comparison of approaches to deriving PAI from remote sensed data**

We found substantial differences in PAI values estimated from TLS and DHP and from different TLS processing methods (Figures 3 and 4). Further, differences between TLS methods varied across plot structure, with the greatest differences between methods in plots with high CAI, and therefore high canopy density. Although previous studies have presented TLS as an improvement over DHP due to its independence of illumination and sky conditions during the data acquisition phase, and ability to resolve fine-scale canopy elements and gaps (Calders et al., 2018; Grotti et al., 2020; Zhu et al., 2018), we have shown that there is large variability between TLS processing methods in Mediterranean forests. Rigorous intercomparison of approaches, ideally using standard benchmarking TLS datasets, and destructive sampling, would improve trust and reliability of TLS algorithms.

We found the *LiDAR Pulse* method (Jupp et al., 2008) to have the best agreement with DHP for both whole plot and single scan PAI estimates. In contrast to previous studies comparing $PAI_{TLS}$ with $PAI_{DHP}$ (Calders et al., 2018; Grotti et al., 2020; Woodgate et al., 2015), we found that the *LiDAR Pulse* and *2D Intensity Image* methods underestimated PAI compared to DHP, except at very low PAI values ($PAI_{TLS} < 0.5$). Quantification of PAI from DHP may introduce additional sources of error, for example, its relatively lower resolution compared to TLS could lead to mixed pixels that have a greater chance of misclassification of sky as vegetation (Jonckheere et al., 2004). This effect could be enhanced in a Mediterranean forest as trees in drier climates tend to have smaller leaves (Peppe et al., 2011), leading to more small canopy gaps that TLS may resolve where DHP cannot. Further, although we took steps to reduce the error introduced at DHP data acquisition and processing steps, including using automatic thresholding and collecting images with multiple exposures, DHP processing requires both model and user assumptions that can impact results. For example, $PAI_{DHP}$ estimates are highly sensitive to camera exposure; increasing one stop of exposure can result in $3 - 28\%$ difference in PAI and use of automatic exposure can result in up to 70% error (Zhang et al., 2005).

We found the *Voxel-Based* method overestimated PAI values compared to the other methods at the whole plot level. This is likely due to the method's use of co-registered scans, rather than averaged single scan PAI values, since co-registered scans will reduce occlusion effects prevalent in single scan data that could to lead to an underestimation of PAI (Wilkes et al., 2017). The *Voxel-Based* method is, however, sensitive to voxel size (Li et al., 2016), and larger voxels lead to larger PAI estimates as they are unable to capture all of the intricate details of canopy structure; we chose a voxel size of 0.05 m to match the minimum distance between points in our downsampled dataset. However, the *Voxel-Based* method is a memory intensive approach to calculating PAI, and smaller voxels have higher memory requirements. We picked this data resolution, and therefore voxel size, to balance the need to capture fine-scale canopy details against memory requirements for running the method on many large plot point clouds. Voxel size could have been chosen based on estimates' match to DHP, but this would assume (1) that DHP estimates are most accurate, and (2) that DHP data are always available, limiting the wider applicability of our findings. Understanding which method is over- or underestimating would require a destructively sampled dataset for validation, which was not possible for this study (or most ecosystems). However, other studies using voxel approaches have found that although these produce high LAI values for individual trees, these are underestimates compared with destructive samples (Li et al., 2016). Regardless, PAI and LAI estimates

using a *Voxel-Based* approach are highly dependent on voxel size (Li et al., 2016), and future work should test the influence of voxel size on PAI estimates, using destructive samples in a range of environments.

The relationship between the *LiDAR Pulse* method and TLS derived CAI had the highest $R^2$, demonstrating that the method is well suited to measuring PAI across the range of plot CAI values used in this study. Although the *2D Intensity Image* method can tackle the significant challenges presented by edge effects and partial beam interceptions, particularly present in phase-shift systems (Grotti et al., 2020), our results suggest this method has a lower performance ability, with saturation occurring sooner than all other methods in dense forests (Figures 3 and 4). The *2D Intensity Image* method uses the same raw single scan data as the *LiDAR Pulse* method, so the better performance from the latter is likely due to the method's use of vertically resolved gap fraction; both the *LiDAR Pulse* method and *Voxel-Based* method account for the vertical structure of the canopy by summing vertical slices through the canopy.

**4.2 α variation between species and plot**

We used the *Voxel-Based* method to investigate individual tree α variation between species and across structure, as this was the only approach we compared that could be applied to single tree point clouds which are leaf-wood separated. We found α values obtained were within the range of values obtained from destructive approaches (0.1 – 0.6, Gower et al., 1997). The drought and shade intolerant *P. nigra* showed stronger variability in α across plots (higher ICC value, Table B2) than other species, suggesting its wood – leaf ratio may be more sensitive to site factors. However, as the plots measured in this study vary in both abiotic conditions (altitude, aspect, slope, wetness) as well as species composition, stem density and canopy cover, there may be other drivers of variation in α values.

We found some evidence that species with higher drought tolerance had higher α values (Figure 5; Table B1), however, confidence intervals were wide, suggesting a weak relationship. There is evidence that trees that tolerate water limited environments have a lower leaf area (Battaglia et al., 1998; Mencuccini and Grace, 1995), so higher α values may reflect maintenance of homeostasis of leaf water use through adjustment of wood to leaf area ratio (Carter and White, 2009; Gazal et al., 2006). The potential for a tree to lose water is mostly regulated through leaf traits including stomatal conductance and leaf area, and both stand (Battaglia et al., 1998; Specht and Specht, 1989) and individual tree (Mencuccini, 2003) water use have been found to scale linearly with LAI, with drought often mitigated through leaf shedding (López et al., 2021).

**4.3 Tree stature and stand density drives α variation**

Although species had a weak relationship with α, tree height and plot CAI had a statistically significant relationship with α ($p < 0.001$ – $p < 0.05$) for all species, showing the importance of local stand structure on leaf and woody allocation. We found that α scaled negatively with height for all species apart from *P. sylvestris,* suggesting that in this environment, taller trees generally have a lower proportion of wood to plant area index than shorter ones. *P. sylvestris,* which is at the edge of its geographical range and physiological limits (Castro-Díez et al., 1997; Owen et al., 2021), showed no significant relationship between height and α. We found that α scaled positively with plot level CAI for all species apart from *P. sylvestris,* that is, trees growing in denser plots have a higher α. This supports theory that trees growing in dense forests are competing for resources, reducing individual tree leaf area (Jump et al., 2017). The negative relationships between height and α and positive relationships

between CAI and α relationships in our model suggest that trees may initially invest in vertical growth to reach the canopy level, and once there invest in lateral growth, with more leaf area, to increase light capture. This supports theory that trees grow to outcompete neighbouring individuals for light capture (Purves and Pacala, 2008) and evidence that both lateral growth and LAI are reduced beneath closed canopies (Beaudet and Messier, 1998; Canham, 1988).

Wood may be harder to accurately classify than leaves in TLS data (Vicari et al., 2019), resulting in a higher occurrence of false positives in wood clouds, potentially leading to an overestimation in WAI, and therefore underestimation of α, especially in trees with small leaves which are prevalent in dry, Mediterranean environments (Peppe et al., 2011). The problem of misclassification will increase in taller trees due to TLS beam divergence, occlusion and larger beam footprint at further distances (Vicari et al., 2019), suggesting that WAI overestimation could be more pronounced in tall trees. Although our dense scanning strategy (Owen et al., 2021) was designed to mitigate some of these effects, these effects mean our findings may underestimate the slope of the negative relationship between α and tree height. Conversely, the increasing leaf-to-wood ratio could potentially be explained by a greater number of empty voxels caused by occlusion in large trees. However, we took significant steps to reduce occlusion, employing a 10 m scanning strategy that was developed in a dense tropical forest (Wilkes et al., 2017).

**4.4 Correcting for non-photosynthetic elements in LAI estimates using TLS**

The value of TLS data to estimate individual tree PAI, WAI and subsequently α, demonstrates their potential to corrective factors for non-photosynthetic components in ground based remote sensing measurements of LAI. Properly correcting for WAI in LAI estimates is of global importance as small errors in ground based measurements propagate through to large scale satellite observations generating large errors in global vegetation models (Calders et al., 2018). The work presented here provides a foundation for future work combining multi-source and multi-scale remote sensing datasets to correct largescale LAI products. Our results echo others' in finding that the prevalence of woody material in the tree canopy, and therefore α, is dynamic and varies by species as well as senescence, crown health and, in the case of deciduous forests, leaf phenology (Gower et al., 1999). The use of single α value in a plot or region (Olivas et al., 2013; Woodgate et al., 2016), invariant of species, size and forest structure, to convert PAI to LAI is therefore problematic (Niu et al., 2021). Our study demonstrates the importance of taking species mix and structural variation into account when correcting for non-photosynthetic material in ground-based LAI estimates.

**5. Conclusions**

We tested three methods for estimating PAI using Terrestrial Laser Scanning data and compared these against traditional DHP measurements. We found large variation between PAI values estimated from each TLS method and DHP, demonstrating that care should be taken when deriving PAI from ground based remote sensing methods. Although the *LiDAR Pulse* method was found to have the best agreement with both single scan and whole plot PAI values measured by DHP, the *Voxel-Based* method allowed separate analysis of the key metric used to correct for the effect of WAI in LAI measurements, α, in individual trees. We recommend the *LiDAR Pulse* method as a fast and effective method for PAI estimation independent of illumination conditions. Whilst the *Voxel-Based* method may be used to analyse individual tree α and determine ecological drivers of variation, work remains to

determine the validity of these approaches, in particular correct voxel size choice. We found that α varies by species, height and stand density, showing the importance of accurately correcting for WAI on the individual tree level and the utility of TLS to do so.

The variation in our results for the different methods used to derive PAI from TLS data show that there is some way to go before TLS derived vegetation indices can be interpreted as robust and reliable. Validation using destructive samples and further intercomparison studies of methods are needed to demonstrate the advantages of TLS, and use of benchmarking datasets should be standard. DHP is a faster, cheaper and more widely accessible method for PAI estimation, and while TLS promises to alleviate potential bias in DHP estimates, results are highly methods dependent. Our results demonstrate the challenges that stand in the way of large scale adoption of TLS for vegetation indices monitoring.

## 6. Appendices

### 6.1 Appendix A

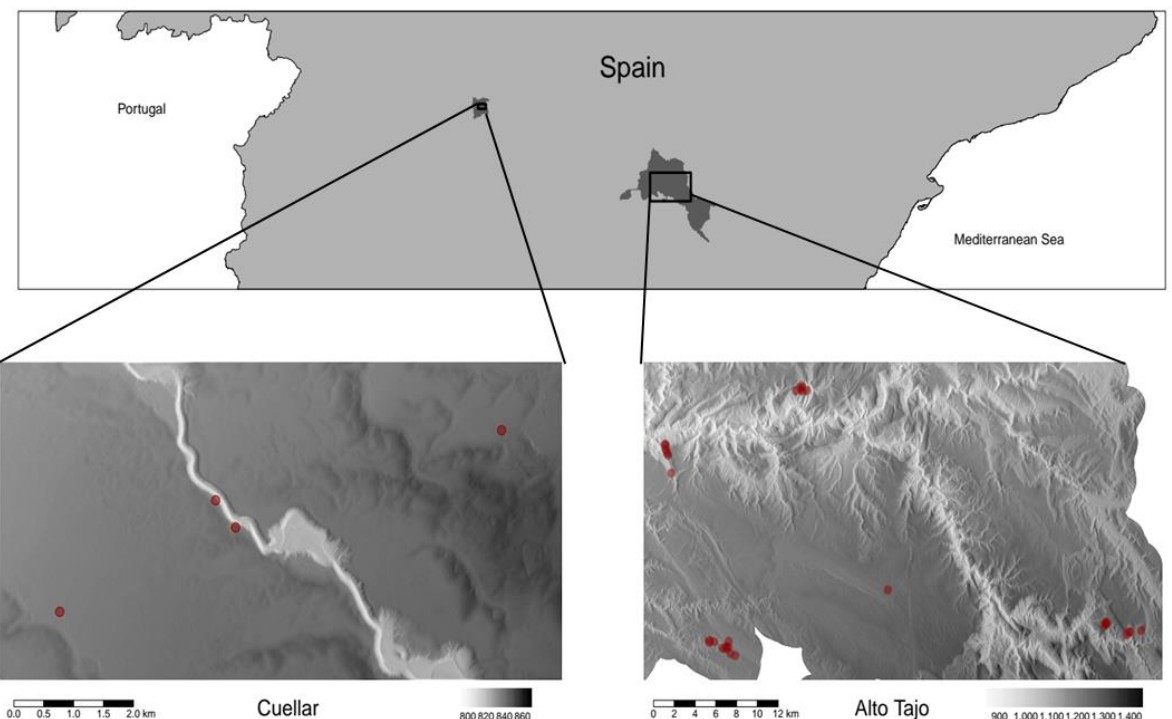

**Figure A1: Map of plot locations within two field sites in central Spain (Cuellar, left and Alto Tajo, right). Red points show plot locations on high-resolution digital terrain models enhanced with hillshading shown in greyscale (Owen., 2021).**

### 6.2 Appendix B

**Table B1: species – α linear mixed model (equation 1) showing relationship between tree species and α for all 2472 individual trees. Species are listed from low – high drought tolerance, with the exception of *P. pinaster*, for which drought tolerance index has not been calculated in the literature. 95% CI are the 95% confidence intervals.**

| Species | *a* (eq. 1) | 95% CI |
|---|---|---|
| *P. sylvestris* | 0.144 | 0.131, 0.158 |
| *P. nigra* | 0.138 | 0.127, 0.149 |
| *Q. faginea* | 0.149 | 0.140, 0.157 |
| *Q. ilex* | 0.155 | 0.146, 0.166 |
| *P. pinaster* | 0.168 | 0.145, 0.192 |

**Table B2: height – α linear mixed models for each species (equation 2) showing relationship between tree height and plot CAI and α for all 2472 individual trees. Species are listed from low – high estimated α. Significance codes: p < 0.001 '\*\*\*'; p < 0.01 '\*\*'; p < 0.05 '\*'; not significant 'ns'. 95% CI are the 95% confidence intervals and ICC is the intra-class correlation coefficient.**

| Species | *b* (eq. 2) (95% CI) | *c* (eq. 2) (95% CI) | ICC |
|---|---|---|---|
| *P. sylvestris* | $-0.002^{ns}$ (-0.004, 0.000) | $0.134^{ns}$ (0.010 0.259) | 0.151 |
| *P. nigra* | -0.005\*\*\* (-0.006, -0.004) | 0.164\*\* (0.063, 0.263) | 0.211 |
| *Q. faginea* | -0.008\*\*\* (-0.010, -0.007) | 0.058\* (0.016, 0.101) | 0.060 |
| *Q. ilex* | -0.015\*\*\* (-0.020, -0.011) | 0.113\*\* (0.050, 0.179) | 0.070 |
| *P. pinaster* | -0.006\*\*\* (-0.008, -0.004) | 0.317\* (0.177, 0.453) | 0.036 |

### 6.3 Appendix C

$$WAI = m_{species} \quad \text{(C1)}$$

$$WAI = m\,height + b \quad \text{(C2)}$$

$$WAI = m\,CAI + b \quad \text{(C3)}$$

$$WAI = m\,PAI + b \quad \text{(C4)}$$

Where WAI is the wood area index, *species, height, CAI* and *PAI* are the tree species, tree height, crown area index of the plot in which the tree is growing and tree plant area index respectively and *m* and *b* are parameters to be fit.

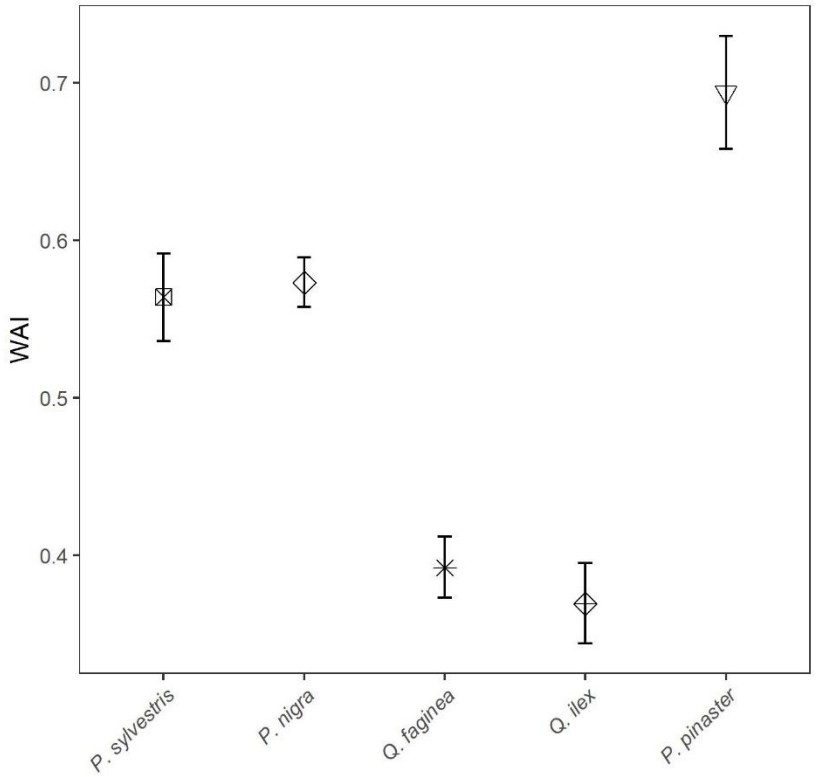

486

**Figure C2: Linear model derived WAI values (m, equation C1) for all 2472 individual trees of species *P. sylvestris, P. nigra, Q. faginea, Q. ilex* and *P. pinaster*. Error bars represent 95% confidence intervals. Species are listed from low – high drought tolerance, with the exception of *P. pinaster*, for which drought tolerance index has not been calculated in the literature. Between-species differences in WAI are likely primarily driven by differences in average tree height.**

**Table C3: Linear model (equation C1) showing relationship between tree species and WAI for all 2471 individual trees. Significance codes: $p < 0.001$ '***'; $p < 0.01$ '**'; $p < 0.05$ '*'; not significant 'ns'. 95% CI are the 95% confidence intervals.**

| Species | $m$ (eq. 1) | 95% CI |
|---|---|---|
| *P.nigra* | 0.57*** | 0.56, 0.59 |
| *P. pinaster* | 0.69*** | 0.66, 0.73 |
| *P. sylvestris* | 0.56(ns) | 0.54, 0.59 |
| *Q. faginea* | 0.39*** | 0.37, 0.41 |
| *Q. ilex* | 0.37*** | 0.34, 0.39 |

494

**Table C4: Linear models (equations C2, C3, C4) predicting WAI as a function of tree height, CAI (density) and PAI Significance codes: $p < 0.001$ '***'; $p < 0.01$ '**'; $p < 0.05$ '*'; not significant 'ns'. 95% CI are the 95% confidence intervals.**

| | $m$ (eq. 2, 3, 4) (95% CI) | $b$ (eq. 2, 3, 4) (95% CI) | $R^2$ |
|---|---|---|---|
| Tree Height | 0.024*** (0.023, 0.026) | 0.27*** (0.25, 0.28) | 0.27 |
| CAI | 0.390*** (0.336, 0.443) | 0.29*** (0.26, 0.31) | 0.78 |
| PAI | 0.112*** (0.106, 0.118) | 0.12*** (0.10, 0.14) | 0.35 |

498

## 7. Code availability

See https://github.com/will-flynn/tls_dhp_pai.git for all processing and modelling code.

**8. Data availability**

See Owen et al. (2022) for individual segmented tree data and Flynn et al. (2023) for thresholded DHP images.

**9. Author contribution**

All authors designed the study. HJFO and WRMF collected and processed TLS and DHP data; WRMF performed formal analysis with guidance from all authors. WRMF led the writing with input from all authors. All authors contributed critically to drafts and gave final approval for publication.

**10. Competing interests**

The authors declare that they have no conflict of interest.

**11. Acknowledgements**

WRMF was funded through a London NERC DTP PhD studentship. ERL, HJFO and SWDG were funded through the UKRI Future Leaders Fellowship awarded to ERL (MR/T019832/1).

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
