# Peer review of "Quantifying vegetation indices using Terrestrial Laser Scanning: methodological complexities and ecological insights from a Mediterranean forest"

_EGUsphere, 2022_

## Referee Comment (RC2)

I think this work is meaningful for monitoring and modeling carbon storage and uptake of Mediterranean tree species. Although they extracted canopy structure parameters using existing methods, evaluating the WAI of different tree species is valuable for relevant ecological and forestry research. I think this article's first and foremost improvement point is that the research goal is not clear and in-depth enough. The presentation of the results is not complete, which makes it difficult for readers to capture their needed information, such as the WAI variation trend (functions) among 5 tree species related to their height and density.

Authors think the TLS-based canopy metrics are not yet reliably calibrated and validated to the extent they are ready to replace traditional approaches for large-scale monitoring of PAI and LAI. In most studies, TLS data only depicts the tree structure at the plot- and single-tree scale. While using other remote sensing technologies cannot depict the fine structure features of the internal canopy and separating woody and foliage materials. Referring to the DHP results, authors evaluated the error of WAI and PAI analyzed by point clouds. I would like to know if the authors use the TLS data to improve the LAI evaluation accuracy. TLS can support assessing the single tree and plot-level WAI more accurately. More importantly, whether the WAI of different Mediterranean trees has similarities between the same species, as well as providing specific information (maybe list in thematic tables to show the relationship among species, tree height, density, and PAI), will make readers benefit greatly. I think the measured data of this study can support this research goal, while they are not fully presented in the current edition.

In addition, the presentation of the results is incomplete. I did not find the location, site conditions and tree species appearance of the measured plots shown in the manuscript. The segmentation results of different tree species and the statistical information on PAI and WAI of trees grown in different site conditions were also not provided. In addition, critical mathematical functions and quantitative conclusions are also lacking in the current edition.

I suggest authors reconsider whether it is necessary to study the CAI. This parameter can be easily analyzed using remote sensing images without using TLS.

Furthermore, is it applicable to use a fixed voxel size when analyzing WAI? After all, different tree species have various canopy shapes and branch structure features. Adaptive adjusting the voxel size according to the point cloud density and the branch distribution trend may be more reasonable.

Optimizing the TLS-based WAI assessment methods, summarizing the regulation of interspecific WAI variation, and using these rules to improve the LAI assessment will make this article more attractive to better support research in related fields. **The following are some detailed points.** I hope they will help improve the current edition.

➢ I suggest authors clarify their research goal in the initial section of the manuscript. As a reader, I am more interested in how to use TLS to analyze WAI. However, authors did not briefly introduce the WAI extraction methods in the abstract but focused on comparing point cloud extraction methods of PAI and LAI.

➢ They focused on the wood to total plant area ($\alpha$). I wonder if it is feasible to measure the plant area because of the occlusion effect during scanning. TLS may be more suitable for analyzing WAI.

➢ Section 1.3 It will be more interesting to add some research topics on integrating the fine-scale WAI (or $\alpha$) assessed based on TLS to correct the large-scale LAI extracted from the multi-source remote sensing images. Based on the high-quality field dataset, it should be feasible to use this research in optimizing the large-scale LAI distribution evaluation.

➢ In Sections 2.1 and 2.2, the location map of study plots and some images showing the scene of plots should be provided. The pictures of tree species also need to be added to show their phenotypic characteristics, which is beneficial to evaluate their drought tolerance (L323-324).

➢ L 191 When setting this threshold (> 0 points) to identify the filled voxels, did you filter noisy points out from the tree TLS datasets? It is not easy to identify and filter all noise in TLS data. I am worried the noise would lead to a lower $P_{gap}$ and cause inaccurate LAI and PAI.

➢ L203-204 Some structure features of woody and foliage materials can be analyzed based on the pointset-, height bin-, and patch-based models. Please

revise this sentence.

➢ L206 The principle of TLS segmentation methods needs to be briefly introduced before the voxelization step. It is beneficial to improve the readability of the manuscript.

➢ L216 How to analyze the WAI after voxelizing woody point clouds? Some details should be introduced, which is key to calculating ɑ.

➢ L225 Why explore the relationship between PAI and CAI in this study? The CAI assessment seems to deviate from the research topic, as it is not highly related to LAI and WAI but to the crown projection area, except the canopy gap area. Moreover, using images for CAI analysis is sufficient.

➢ L245 As shown in Figure 3, PAI estimated using the LiDAR Pulse method more strongly agreed with DHP PAI than the Intensity Image method. However, I found their correlation ($R^2$) is not particularly significant.

➢ L248 Please carefully recheck the description of the results is correct according to Figure 3. As shown in Figure 3a, the Pulse-based method overestimates the PAI, while the intensity-based method underestimates the PAI.

➢ L264 You did not label Voxel-Based PAI in Figure 3. Do you mean the TLS PAI?

➢ L269 Maybe you did not set a suitable threshold when defining blank voxels. Merely my speculation!

➢ L282 and 257 You forget to mark the 1:1 dash line in these figures.

➢ Although authors used the published woody-and-foliage separation methods, it is necessary to display some examples of TLS separation results scanned from diverse plots grown with different species. Due to the lack of validation data, it may be challenging to evaluate the segmentation accuracy. However, presenting the separation results is still available to support visual evaluation.

➢ It is not easy to accurately separate the branch and leaf point clouds of trees except those of broadleaf. More importantly, I am worried about whether it is applicable to use the same voxel size to calculate the WAI of different tree species, which is crucial to the conclusion.

➢ L294-297 These sentences are not clear. How to assess tree-specific drought

tolerance? You would better add some description about its evaluation methods and list the metrics to evaluate the drought tolerance of different tree species in this figure and the related references.

➤ In section 4.1, why did you discuss the plot-scale CAI variation? The topic of this section is comparing diverse approaches to deriving PAI.

➤ The title of Section 4.2 is a phenomenon that you need to analyze. Sections 4.2 and 4.3 still belong to Section 4.1 to discuss the LiDAR-extracted metrics with that of DHP.

➤ L320 According to the field data and Figure 3, what is a very low PAI value? Providing a quantitative indicator will significantly improve the manuscript's readability than using adjective words.

➤ L348 The highest $R^2$ does not show a strong correlation.

➤ L374 This sentence is not clear. "Although species explain some variation in α, tree height and plot CAI were stronger predictors for all species…." According to the principle of these parameters, it is hard for me to agree that CAI and WAI have a strong correlation.

➤ L390-392 This is an interesting point. I prefer you to provide some figures and statistical information to prove your finding, especially in different plots with variable growing patterns (growing density, CAI, and WAI related to the tree species, as you mentioned in the Conclusion section). It is beneficial to deepen this study topic.

➤ L 398 I agree that correcting WAI can improve the LAI assessment. The TLS-extracted data can support calibrating LAI based on WAI and PAI. The WAI may be similar among single trees of the same tree species. According to your results, the WAI shows a more evident relationship to tree height and stand density. I think the assessed WAI and plot-level PAI can be used to correct regional LAI for the plot or large-scale forests that were growing with limited tree species.

**Some text errors that needed to be corrected are listed as follows:**

● Do not use an abbreviation in the title of your manuscript, as many readers in other fields do not know the meaning of TLS.

- I suggest authors unify the reference format throughout their manuscript. Different citation formats appear in the same paragraph may confuse readers.

- L135   What are FunDIV plots?

- L142   I do not understand "altitudinal gradient 840 – 1400 m.a.s.l.".

- L167   compare – >  compared

- L169 and 180    Please note the font size of the subscript in the Pgap. This abbreviation can also be used in line 162.

- L176 Please add a comma to this sentence.

- L199 Where are the solid black voxels in Figure 2?

- L209 wood only point clouds?

- L210 TLSeparation classifies points as leaf or wood? This sentence is not clear.

- L219 TLS PAI and DHP PAI? (Using $PAI_{TLS}$ and $PAI_{DHP}$ instead)

- L234 Please add a comma to this sentence.

- L246-248, L264-265 You can mark these metrics in the insets of Figure 3.

- Points in Figure 3 can be denoted as different marks or colors, such as circles or crosses, red or blue, to make this chart clearer (like the style of Figure 4).

- L262 Please unify the term throughout the manuscript. I think TLS whole plot PAI means TLS PAI($PAI_{TLS}$).

- L274-276 You would better mark these metrics in the subfigures of Figure 4.

- In Figures 3 and 4, please delete the unit of PAI. The PAI, LAI and WAI are all ratio-type parameters (no need to denote unit).

- L318 TLS – DHP comparisons?

- In this article, authors used lots of open-source software to support their analysis. I suggest they list all applicable packages and download links to make readers easy to use these tools.

- Please carefully check the format of all references according to the manuscript preparation guidelines and the latest published papers in Biogeosciences. The current reference format needs to be optimized.

---

## Author Comment (AC1)

We thank the reviewer for their comments, which we have discussed in the responses below and which we believe have significantly improved the manuscript.

**1.1 Regarding the methods, I am afraid that the authors have overlooked a significant part of the recent scientific literature on the subject. The voxel-based approach which is presented, tested and discussed in the manuscript is not an undisputed reference method and has a number of known drawbacks.**

We agree with the reviewer's view that a "best approach" to voxel-based PAI estimation remains contested. However, the aim of this paper is not to evaluate all possible voxel methods, but rather use a method with broad applicability to multiple TLS configurations. We chose the voxel-based method used in this study for clear reasons. First, we wanted to use full plots and segmented trees, so methods developed with single scans were inappropriate. Further, many radiative transfer methods require information on scanner location and beam direction limiting use to single scans or individual trees with known scan locations around them – not available in many TLS datasets. In addition, our preference where possibly is to use methods that have been thoroughly and independently validated – in this case the voxel method chosen has been validated with destructive sampling. Finally, we only use methods that were open source and easily reproducible, excluding many insufficiently documented, GUI-based or proprietary approaches. Further discussion of our choice is given in our answer to the following comment.

We note that efforts to move towards a best practice consensus are building within the community requiring a dedicated effort and we believe our study provides direction for ways forward. We also highlight that methods should be compared across sensor and forest types to draw robust conclusions. As the data used in this study are published, we would be delighted to see further exploration of this topic making use of different voxelisation schemes but see the testing of these different methods to be beyond the scope of this study.

**1.2 L103-106 the authors briefly mention that there are different approaches to voxel-based estimation of PAI/LAI and they opt for one that treats elementary voxels as either empty or full (opaque). Unfortunately, there is no obvious justification for such a choice being made. The emerging consensus in the recent literature seems to be in favour of what the authors refer to as "simulating radiative transfer within each cube". One significant advantage being that the laser scanning geometry is considered, and hence the variable sampling intensity and occlusion effects on PAI estimated can be accounted for.**

While we appreciate the reviewer's comment that significant recent progress has been made in the field, our view is that a consensus on best approach is yet to be reached and the approach proposed by the reviewer is still contested (and please see the proceeding comment). In particular, there is increasing recognition that voxel size significantly influences PAI/WAI/LAI estimates, and many methods do not provide clear guidance on how to deal with this. For example, You et al., (2022), published after the submission of this manuscript, argue that voxel-based methods are highly sensitive to voxel size and present a morphology-based method to obtain LAI from the surface area on envelope fitting to extracted leaf points. A key benefit of the voxel-based approach used in this study is the clear justification for matching voxel size to point cloud resolution, as evaluated in Li et al., (2016) and validated using destructive samples. Using a radiative transfer approach PAI estimates are highly unstable over varying voxel sizes and there is no clear guidance from the literature on how to choose the correct one. Evaluating the many potential methods for calculating LAI from TLS data are well beyond the scope of this study, however, we hope that the work presented here will contribute towards a future consensus in the field.

As discussed in our response to reviewer 2 (comment 2.9), we have amended our discussion of voxel size in section 1.3 to reflect the debate around voxel size choice, L108-113:

*"However, PAI estimates derived using the voxel method are highly dependent on voxel size (Calders et al., 2020). Using a radiative transfer approach, Béland et al., (2014) demonstrated that voxel size is conditional on canopy clumping, radiative transfer model assumptions and occlusion effects, making a single, fixed choice of voxel size within methods for all datasets impossible. To test various approaches to selecting voxel size using a voxel classification approach, Li et al., (2016) matched voxel size to point cloud resolution, individual tree leaf size, and minimum beam distance and tested against destructive samples, finding that voxel size matched to point cloud resolution had the closest PAI values to destructive samples."*

To clarify our justification for use of a voxel classification approach over a radiative transfer approach, also commented on by reviewer 1, we have added to section 2.4 (L199-201):

*"We chose a voxel classification approach as this method is widely applicable to a range of TLS systems and levels of processing as well as providing explicit guidance on voxel size selection, which is known to impact derived PAI estimates (Li et al., 2016)."*

**1.3 In addition, degrading point cloud resolution down to the voxels resolution is likely to degrade the quality of point cloud segmentation into leaf and wood as well as the PAI estimates.**

Downsampling is a critical step in *treeseg* (Burt et al., 2019) to handle computational loads associated with segmenting point clouds. We thank the reviewer for drawing attention to our lack of clarity over the justification for down sampling data and have added to L220-224:

*"We used individual tree point clouds downsampled to 0.05 m, to aid computation time, and segmented using the automated tree segmentation program treeseg (Burt et al., 2019), implemented in C++, by Owen et al., (2021) for that study. Individual segmented tree data are available in Owen et al., (2022)."*

The requirement of individual tree point clouds in TLSeparation means downsampled individual tree point clouds are necessary without upscaling the resolution of individually segmented trees. The scale of this study (2472 trees, 33 plots) means using a raw data resolution is computationally impracticable and consequently, downsampling is common practice in studies using large datasets of individual tree point clouds. We believe choosing a *knn* based on the point cloud resolution is a robust approach to optimising wood leaf separation under the constraints associated with large datasets. We explain the *knn* in L229-231 of the submitted manuscript:

*"The knn of each iteration is directly dependent on point cloud density, since high density point clouds will require higher a knn (Vicari et al., 2019). We used the utility package in TLSeparation to automatically detect the optimum knn for each tree point cloud."*

We chose point cloud resolution as a trade-off between computational demands while retaining the structural information contained in each tree. We then matched voxel size to point cloud resolution rather than down sampling point cloud resolution to desired voxel size. This is in line with recommendations for the method we chose; many voxel-based methods provide no clear guidance on this. We thank the reviewer for drawing attention to our lack of clarity here and have added to L195-199:

*To calculate PAI using the Voxel-Based method, we followed a voxel classification approach* (Hosoi and Omasa, 2006)*, downsampling the point cloud to 0.05 m to aid computation time and matching the voxel size to the resolution of the point cloud, following Li et al., (2016), who showed that matching the voxel size to the point cloud point to point minimum distance (resolution) increases accuracy as small canopy gaps are not included in voxels classified as vegetation.*

**1.4 For these reasons, I believe the conclusions drawn are not well grounded. The general conclusion that "Our results demonstrate the challenges that stand in the way of large scale adoption of TLS for vegetation indices monitoring" which refers to the large discrepancies observed between methods in their study contradicts recent papers such as (Béland and Kobayashi, 2021; Nguyen et al., 2022). Obviously, there are still challenges to address but this study does not seem to identify the real caveats associated with the use of TLS in vegetation studies.**

From the cited literature we assume the reviewer is referring to (1) voxel size and (2) occlusion.

Regarding voxel size, we agree that there is a major problem in choosing voxel size with little consensus on how to choose the correct one for a range or forest types and ecosystems (see responses above). Separate analysis performed within our group shows unstable results over a range of voxel sizes using a radiative transfer approach, with a wide range of derived indices for one scan across relatively small variation in voxel size, suggesting high model sensitivity to this input parameter. The method we chose matches the voxel size to the resolution of the point cloud, and while the reviewer has pointed out there are "a number of known drawbacks" with this method, we feel that in the absence of well justified methods this is a pragmatic approach to accurately choosing the correct voxel size, and has been validated with destructively sampled data.

Regarding occlusion, Béland and Kobayashi, (2021) have chosen a very dense scanning density (5 m between scans), which is impractical in large-scale forest plots, and greater than the suggested scanning density in Wilkes et al., (2017), making such a dataset rarely available. Béland and Kobayashi, (2021) also suggest site specificity for their results, focusing on broadleaf trees, limiting the applicability of findings to our mixed Mediterranean forest.

Finally, the conclusion reached in our paper that "challenges stand in the way of large scale adoption of TLS" are drawn from a comparison of three TLS methods with conventional DHP. Neither papers cited (Béland and Kobayashi, 2021; Nguyen et al., 2022) test a voxel-based method against other widely used TLS PAI derivation methods (e.g. LiDAR pulse, 2D intensity image) and DHP, rather they are focused entirely on a voxel-based approach. We therefore argue the novelty of our findings and believe they do not contradict these papers.

**1.5 Both theory and algorithms have advanced significantly in recent years and convergent approaches to PAI/LAI estimates from lidar (both TLS and ALS) are emerging. Maybe the authors will want to check the following references**

We thank the reviewer for the references provided, however, argue that our dataset is significantly different from the data used in these studies. Methods suggested by the reviewer have been developed with individually scanned trees or branches (e.g. Béland et al., 2011; Soma et al., 2018), or with simulated data (e.g. Grau et al., 2017; Pimont et al., 2019, 2018; Soma et al., 2020). Individually scanned trees or branches can be scanned with a set of known scan positions allowing the precise location, distance, and beam angle from the scanner to be derived. Further, Béland and Kobayashi, (2021) focused on broadleaf trees functionally and physiologically different to those in our study, used a prohibitively dense scanning strategy (5 m), and lack validation from destructive sampling. Our dataset comprises 2472 trees scanned from 528 locations. To derive point-level information containing scanner location and beam angle would add significant complexity and computational load to the study. While an important question, understanding the necessity for this added complexity is beyond the scope of this paper.

As stated in the proposed manuscript L26-28: *"Our findings highlight the value of TLS data to improve fundamental understanding of tree form and function, but also the importance of rigorous testing of TLS data processing methods at a time when new approaches are being rapidly developed.",* we argue that the purpose of this paper is not to evaluate the latest methods, rather to take a step back and test existing methodologies with a large dataset.

**1.6 the authors refer to Beland et al. 2014 when noting the potential role of voxel size in the voxel-based approach, but that paper uses a voxel-based approach which is not the one used by the authors**

Thank you for pointing out this inappropriate reference to Béland et al., (2014); we apologise for this mistake and have corrected it in L346, changing the reference to Li et al., (2016) who found voxel size to have significant effect on PAI estimates using the same voxel-based approach used in the study.

**1.7 Regarding the ecological insights, the clearest result seems to be that the alpha parameter (WAI/PAI) decreases with tree size (figure 6). The interpretation of what may appear as a paradox is largely speculative. It is interpreted as the result of competition but no data supporting this is presented. One might have tried to explore how alpha evolved in relation to the local competition index for instance.**

We thank the reviewer for the suggestion of exploring how alpha evolves with local competition, which is a key finding of this paper that we have not sufficiently highlighted. Figure 6b shows how alpha changes in relation to plot-level crown area index (CAI), a measure of the plot area covered by tree crown, and one that we have used as an indicative measure of local competition.

To clarify the use of CAI as indicative measure of local competition, we have changed the wording in L246-249 to state:

*"To further understand observed drivers of variance in PAI, we tested the relationship between PAI and TLS estimated whole plot crown area index, CAI, calculated as the sum of projected crown area divided by the plot area* (Owen et al., 2021)*, and a proxy measure of stand density and local competition* (Caspersen et al., 2011; Coomes et al., 2012)*, using SMA."*

**1.8 There is abundant literature (and theoretical arguments) that indicate that LeafToWood biomass ratio of trees growing in stands will tend to decrease with size (Bartelink, 1997; Forrester et al., 2017; Mensah et al., 2016). In the present study, the WoodToLeaf area ratio is found to decrease with tree size (for the four species for which there is a significant trend in figure 6). This could be an artefact as the authors point out (l. 387-394). The issue might indeed have to do with the leaf/wood filtering.**

We agree with the reviewer that there is abundant literature that argue leaf to wood biomass ratio will tend to decrease with size, however, the literature cited by the reviewer differ fundamentally from our study in ways that may explain differences in results. For example, the focus species, *Fagus sylvatica* in Bartelink, (1997) is functionally different to species analysed in this study; Forrester et al., (2017) evaluate leaf biomass rather than wood to plant ratio and Mensah et al., (2016) omit correction for competition in their models while also excluding the largest trees from the study possibly introducing bias. This means that the arguments presented may not hold in our dataset measured in a mixed Mediterranean forest.

We agree that wood to plant ratio could be influenced by an artifact of wood – leaf classification, and have elaborated on this point in L418-425 of the proposed manuscript: "*Wood may be harder to accurately classify than leaves in TLS data* (Vicari et al., 2019)*, resulting in a higher occurrence of false positives in wood clouds, potentially leading to an overestimation in WAI, and therefore underestimation of α, especially in trees with small leaves which are prevalent in dry, Mediterranean environments* (Peppe et al., 2011). *The problem of misclassification will increase in taller trees due to TLS beam divergence, occlusion and larger beam footprint at further distances* (Vicari et al., 2019)*, suggesting that WAI overestimation could be more pronounced in tall trees. Although our dense scanning strategy* (Owen et al., 2021) *was designed to mitigate some of these effects, it is possible our findings could underestimate the slope of the negative relationship between α and tree height.*". Based on this, we would expect to be underestimating the negative slope of the relationship between alpha and tree height if it was an issue of misclassification.

**1.9 This is also a field where progress has been made in recent years and maybe the authors would want to test alternative algorithms to TLSeparation which might perform better on their data. Some pointers are given below**

We agree that there has been progress in the field of wood – leaf classification, however, we argue most progress has been focused on scaling wood – leaf classification from individual trees to whole scan or plot data (e.g. Krisanski et al., 2021; Wan et al., 2021; Wang, 2020; Wang et al., 2018; Wu et al., 2020) rather than major improvements in the classification framework itself. In the case of LeWoS (Wang et al., 2020), the tool has been tested only with tropical trees and, although, distributed as open-source, is either in the form of Matlab code or a pre-compiled executable, substantially limiting wider applicability. Testing the multitude of available approaches to wood – leaf classification would be invaluable to the field, however, is beyond the scope of this study – not least because such a test should use destructively sampled validation data, which we do not have access to. Here we are interested in using well-established methodology that has been validated with a range of tree types, so based our choice on that criteria.

**1.10 My overall appreciation is that the data collected is very significant and could indeed contribute some new insights in terms of tree/forest ecology but more work is needed prior to publication.**

We thank the reviewer for their comments and appreciate that the reviewer recognises the significance of our data and results. We and are confident that following their and the other reviewer's comments that the manuscript has been significantly enhanced.

**1.11 Reprocessing the TLS data already segmented using an open source freely available code incorporating much of the latest theoretical improvements should not take long. This analysis may profoundly alter the reported results (i.e the large overestimation of PAI with a voxel-based approach and the unexpected negative trend in WAI/PAI with increasing tree size). This may help clarify whether leaf/wood segmentation may be an issue and require further scrutiny or not.**

Whilst additional analyses are always possible, in this case we believe our methodological choices are defensible, and these have been discussed in previous responses. We use wellestablished and tested leaf separation and PAI estimation methods that were tested, in the case of voxel-based method, with destructive samples. The scope of this study is to benchmark the most rigorously available methods, not testing all available methods but taking the conservative approach. Further, all the methods tested in this study are either open source in common programming languages, or, where we have written code this has been made freely available. Not all the methods suggested by the reviewer are open source or easily integrated into automated workflows. We believe that running the analysis again would introduce new, different biases, and don't believe this would enhance manuscript without changing scope.

2006); we chose SMA because we do not have a 'true' validation dataset, so avoid assuming either DHP or any
of the TLS methods produces the most accurate results. For each TLS method, we assessed the relationship with
DHP using the coefficient of determination and RMSE. We chose to compare PAI values rather than WAI or LAI
as each method corrects for non-photosynthetic elements in different ways and would introduce bias, limiting the

[revised manuscript text omitted]

**Appendix C**

$$WAI = m_{species} + b \quad (1)$$

$$WAI = m_{height} + b \quad (2)$$

$$WAI = m_{CAI} + b \quad (3)$$

$$WAI = m_{PAI} + b \quad (4)$$

Where WAI is the wood area index, *species, height, CAI* and PAI are the tree species, tree height, crown area index of the plot in which the tree is growing and tree plant area index respectively and *m* and *b* are parameters to be fit.

[Figure]

**Figure 2: Linear model derived WAI values (m, equation C1) for all 2472 individual trees of species *P. sylvestris, P. nigra, Q. faginea, Q. ilex* and *P. pinaster*. Error bars represent 95% confidence intervals. Species are listed from low – high drought tolerance, with the exception of *P. pinaster*, for which drought tolerance index has not been calculated in the literature.**

**Table 3: Linear model (equation C1) showing relationship between tree species and WAI for all 2471 individual trees. Significance codes: p < 0.001 '***'; p < 0.01 '**'; p < 0.05 '*'; not significant 'ns'**

| Species | *m* (eq. 1) | Std. Error | P value |
|---|---|---|---|
| *P.nigra* | 0.57 | 0.008 | *** |
| *P. pinaster* | 0.69 | 0.018 | |
| *P. sylvestris* | 0.56 | 0.014 | |
| *Q. faginea* | 0.39 | 0.010 | *** |
| *Q. ilex* | 0.37 | 0.013 | *** |

**Table 4: Linear models (equations C2, C3, C4) predicting WAI as a function of tree height, CAI (density) and PAI**
**Significance codes: p < 0.001 '***'; p < 0.01 '**'; p < 0.05 '*'; not significant 'ns'**

|  | $m$ (eq. 2, 3, 4) | $R^2$ | P value |
|---|---|---|---|
| Tree Height | 0.02 | 0.27 | *** |
| CAI | 0.39 | 0.78 | *** |
| PAI | 0.11 | 0.35 | *** |

---

## Author Comment (AC2)

We thank reviewer for acknowledging the impact of this paper and their comments which we have discussed below and think have significantly improved the manuscript.

**2.1 I think this article's first and foremost improvement point is that the research goal is not clear and in-depth enough.**

We apologise for a lack of clarity here, and agree that the twin goals of methodology comparison and ecological insight are not presented in as clear a manner as they could be. We have edited the wording in section 1.3 to improve their readability and enhance the communication of their importance (L132-133):

*"The aims of this study are twofold: the first aim is to compare three TLS methods for estimating PAI with traditional DHP. The second aim of this study is to use TLS to understand drivers of individual tree α variation."*

**2.2 The presentation of the results is not complete, which makes it difficult for readers to capture their needed information, such as the WAI variation trend (functions) among 5 tree species related to their height and density.**

We apologise that our analysis of WAI was not clear to the reviewer. We chose to compare methods based on PAI estimates, and not WAI or LAI, to avoid introducing additional processing steps and complexity and therefore to more directly compare the chosen methodological approaches. Differences in PAI between different TLS and DHP estimates can be attributed to differences in processing approaches, whereas comparison of WAI introduces additional error from separation approaches.

To improve clarity, we have added the following (L241-242):

*"We chose to compare PAI values rather than WAI or LAI as each method corrects for non-photosynthetic elements in different ways and would introduce bias, limiting the ability to directly compare metrics."*

See also our response to comment 2.4 below.

**2.3 Referring to the DHP results, authors evaluated the error of WAI and PAI analyzed by point clouds. I would like to know if the authors use the TLS data to improve the LAI evaluation accuracy. TLS can support assessing the single tree and plot-level WAI more accurately.**

We agree completely that the combination of TLS and DHP might improve analyses, and this has been developed in methods not tested in this paper (e.g. Kamoske et al., 2019).

Here we did not use the two datasets together, preferring instead to retain the ability to compare them as independent estimates of the indices of interest. We note that neither should be viewed as the 'truth', and therefore using them in combination could introduce additional biases that would be challenging to disentangle. Nevertheless, others could use our data to perform the analyses suggested.

**2.4 More importantly, whether the WAI of different Mediterranean trees has similarities between the same species, as well as providing specific information (maybe list in thematic tables to show the relationship among species, tree height, density, and PAI), will make readers benefit greatly. I think the measured data of this study can**

**support this research goal, while they are not fully presented in the current edition.**

Although these are not the focus of this study, we agree that additional information could prove useful to some readers. We thank the reviewer for their suggestion of including WAI analysis, which we think has significantly improved the manuscript. We have added Figure C2 and Tables C3, C4 to the supplementary information and refer to this in the main manuscript, section 3.4 (L320 – 322):

*"To understand drivers of variance in WAI we carried out additional analysis to test the relationship between WAI and species, height, CAI and PAI, and presented these results in Appendix C."*

**2.5 In addition, the presentation of the results is incomplete. I did not find the location, site conditions and tree species appearance of the measured plots shown in the manuscript.**

We apologise for this omission. This information is presented in the cited study Owen et al. (2021), but we have added a detailed site map to the supplementary materials, Figure B1, which is referred to in L145 of the main text.

Please see also our response to comment 2.14

**2.6 The segmentation results of different tree species and the statistical information on PAI and WAI of trees grown in different site conditions were also not provided.**

We thank the reviewer for their comment and apologise for lack of clarity around the segmentation process. Individual tree segmentation was carried out by the authors for a separate study (Owen et al., 2021). We have amended section 2.5 (L218 – 222) to clarify the segmentation process and have signposted (Owen et al., 2021).

Please see also our response to comment 2.17.

**2.7 In addition, critical mathematical functions and quantitative conclusions are also lacking in the current edition.**

We apologise for this lack of completeness. We are not entirely clear to which functions the reviewer refers, but for reasons of clarity and brevity we chose to primarily describe the various processing methods we used rather than repeat their original descriptions, which are extensive within the cited literature. Where equations have been used from other studies, we have cited the original equation number along with the paper in-text (but see response to 2.8 below).

**2.8 I suggest authors reconsider whether it is necessary to study the CAI. This parameter can be easily analyzed using remote sensing images without using TLS.**

In this study we used CAI as a proxy measure of stand density (L248), which was a necessary within our model to both understand and correct for the effect of stand density on wood to plant ratio, α. Controlling for stand density (using CAI as a proxy) is important as trees growing in dense plots have lower water availability per tree (see L75-77). We chose to use CAI as Owen et al., (2021) showed that the metric is also indicative of plot-level competition and the metric accounts for crown overlap which cannot be estimated from imagery in closed forests. Furthermore, Coomes et al., (2012) showed CAI to be better than traditional metrics such as basal area, as it is more intuitive to non-specialists and strongly predicts productivity.

We apologise for the lack of clarity when describing this metric and its intended use in the study, which was also commented on by reviewer 1 (comment 1.7).

We have therefore amended our description of the key metric, CAI, for quantifying stand density and local competition in section 2.6 L245-248:

*"To further understand observed drivers of variance in PAI, we tested the relationship between PAI and TLS estimated whole plot crown area index, CAI, calculated as the sum of projected crown area divided by the plot area* (Owen et al., 2021)*, and a proxy measure of stand density and local competition* (Caspersen et al., 2011; Coomes et al., 2012)*, using SMA."*

**2.9 Furthermore, is it applicable to use a fixed voxel size when analyzing WAI? After all, different tree species have various canopy shapes and branch structure features. Adaptive adjusting the voxel size according to the point cloud density and the branch distribution trend may be more reasonable.**

We thank the reviewer for their comment on voxel size and agree that finding an appropriate voxel size a complex problem (discussed extensively in the response to the other reviewer). We chose the method of voxel classification rather than a radiative transfer approach as it has a definitive method for choosing voxel size based on matching the voxel size to the resolution of the point cloud, which was tested against voxel sizes based on individual tree leaf size, and distance of beam, using destructive samples in Li et al., (2016). Using a radiative transfer approach, the methodology for choosing the "correct" voxel size is not clear, and others' work (and our own additional, unpublished analyses) has shown that estimated PAI values are highly sensitive to voxel size choice.

We have amended our discussion of voxel size in section 1.2 to reflect the contentious debate around voxel size choice, L108-113:

*"However, PAI estimates derived using the voxel method are highly dependent on voxel size* (Calders et al., 2020). *Using a radiative transfer approach, Béland et al., (2014) demonstrated that voxel size is conditional on canopy clumping, radiative transfer model assumptions and occlusion effects, making a single, fixed choice of voxel size within methods for all datasets impossible. To test various approaches to selecting voxel size using a voxel classification approach, Li et al., (2016) matched voxel size to point cloud resolution, individual tree leaf size, and minimum beam distance and tested against destructive samples, finding that voxel size matched to point cloud resolution had the closest PAI values to destructive samples."*

To clarify our justification for use of a voxel classification approach over a radiative transfer approach, also commented on by reviewer 1, we have added to section 2.4 (L197-199):

*"We chose a voxel classification approach as this method is widely applicable to a range of TLS systems and levels of processing as well as providing explicit guidance on voxel size selection, which is known to impact derived PAI estimates* (Li et al., 2016).*"*

**2.10    Optimizing the TLS-based WAI assessment methods, summarizing the regulation of interspecific WAI variation, and using these rules to improve the LAI**

**assessment will make this article more attractive to better support research in related fields.**

We thank the reviewer for their suggestion of including analysis of interspecific WAI variation, which we think is a valuable addition to the paper, and refer to our response to previous comments (2.2, 2.4, 2.11), where we have included these new analyses.

Here, we've focussed on interspecific variation in alpha and PAI, rather than WAI and LAI, but recognise that there would be value in such an additional set of analyses. We agree that developing new methods to correct for WAI in LAI estimates using approaches assessed in this paper would make for exciting work, however we think that to do this well we would require further testing and validation, ideally using destructive samples or multitemporal leaf on/leaf off remote sensing data, which is beyond the scope of this paper.

**The following are some detailed points. I hope they will help improve the current edition.**

**2.11    I suggest authors clarify their research goal in the initial section of the manuscript. As a reader, I am more interested in how to use TLS to analyze WAI. However, authors did not briefly introduce the WAI extraction methods in the abstract but focused on comparing point cloud extraction methods of PAI and LAI.**

We apologise for the lack of clarity in explaining our research goals. As in our response to comment 2.1, we have restated our primary and secondary research goals in section 1.3 (L132-133).

We thank the reviewer for their suggestion of analysing WAI, which we think has significantly improved the manuscript. As for comment 2.4, we have now included this analysis. We have chosen to keep the focus on comparing α, as this value is widely discussed in the literature. We have added a statement to this effect (L234-235):

*"To allow a comparison with existing literature estimating α, (Sea et al., 2011; Woodgate et al., 2016) we focused on α values."*

**2.12    They focused on the wood to total plant area (α). I wonder if it is feasible to measure the plant area because of the occlusion effect during scanning. TLS may be more suitable for analyzing WAI.**

We apologise not clearly stating the reasons for comparing PAI rather than WAI. All remote sensing methods evaluated in this paper (three TLS methods and DHP) more directly measure PAI than WAI or LAI as sensors are measuring the whole plant. Correcting for wood/ leaf to derive WAI/ LAI requires additional processing steps, which vary according to sensor (wood/ leaf separation algorithms for TLS and image masking for DHP, as these systems are not deciduous, and therefore leaf-off scans can't be made), introducing bias and limiting our ability to compare output. We have added a statement to this effect to section 2.6 (L241-242):

*"We chose to compare PAI values rather than WAI or LAI as each method corrects for non-photosynthetic elements in different ways and would introduce bias, limiting the ability to directly compare metrics."*

Please see also our response to comment 2.2

We agree that occlusion is a known problem with TLS data in closed canopy forests, however we have minimised the potential occlusion effects by following a dense scanning strategy following the widely cited Wilkes et al., (2017).

**2.13    Section 1.3 It will be more interesting to add some research topics on integrating the fine-scale WAI (or α) assessed based on TLS to correct the large-scale LAI extracted from the multi-source remote sensing images. Based on the high-quality field dataset, it should be feasible to use this research in optimizing the large-scale LAI distribution evaluation.**

We agree this is an exciting idea and could be the focus of follow-on work. We think that the work presented and, as the reviewer points out, our dataset provides a foundation for a more robust comparison of LAI and new insights from multi-source RS datasets, but that this would be an additional methodological development beyond the scope of our current study.

Following your suggestion, we have added a comment to this effect on L429-430:

*"The work presented here provides a foundation for future work combining multi-source and multi-scale remote sensing datasets to correct largescale LAI products."*

**2.14    In Sections 2.1 and 2.2, the location map of study plots and some images showing the scene of plots should be provided. The pictures of tree species also need to be added to show their phenotypic characteristics, which is beneficial to evaluate their drought tolerance (L323-324).**

We agree with the reviewer that a location map of the study plots would be beneficial to the manuscript and thank them for the suggestion. We have therefore added a new figure, B1 to Appendix B showing the locations of plots within the two field sites, Alto Tajo and Cuellar in central Spain.

We believe that the plots used in this study are well studied and documented in the literature and therefore a detailed description of individual plot characteristics would repeat information already available. We have added signposting to this is section 2.1 (L143-144):

*"We collected TLS and DHP data from 29 plots in Alto Tajo Natural Park (40°41'N 02°03'W; FunDIV plots; see Baeten et al., (2013) for detailed description of plots)"*

The five focus species of this manuscript are widely studied and known species and therefore believe that adding individual images of each species is unnecessary.

**2.15    L 191 When setting this threshold (> 0 points) to identify the filled voxels, did you filter noisy points out from the tree TLS datasets? It is not easy to identify and filter all noise in TLS data. I am worried the noise would lead to a lower $P_{gap}$ and cause inaccurate LAI and PAI.**

We apologise for not making explicit the noise filtering process of our data. We denoised individual-tree point clouds using height dependant statistical filtering as outlined in Owen et al., (2021), and combined individual tree point clouds into whole plots. We have added a statement to this effect to section 2.4 (L199-200):

*"We re-combined individually segmented trees, filtered for noise using a height-dependent statistical filter (see Owen et al., 2021) back into whole plot point clouds"*

While any remaining noise may indeed lead to lower $P_{gap}$, we followed standard processing procedure for this voxel classification method outlined in Hosoi and Omasa, (2006) and tested using destructive samples in Li et al., (2016). Similarly, we followed standard protocol in the published literature for the other two methods (LiDAR Pulse and 2D intensity Image), and therefore consider that our work is a fair representation of each methods' ability to accurately derive PAI and allows a comparison of each methods' merits and drawbacks.

**2.16    L203-204 Some structure features of woody and foliage materials can be analyzed based on the pointset-, height bin-, and patch-based models. Please revise this sentence.**

We apologise for the lack of clarity in this statement, and thank the reviewer for their suggestion. What we meant to say was that the voxel-based approach was the only method compared in this study capable of analysing PAI, WAI and LAI of segmented individual tree point clouds. We have reworded to make this clear and L215-216 now reads:

*"As the only method using multiple co-registered scans, the Voxel-Based method is the only method compared in this study capable of deriving PAI, WAI and LAI of segmented individual tree point clouds."*

**2.17    L206 The principle of TLS segmentation methods needs to be briefly introduced before the voxelization step. It is beneficial to improve the readability of the manuscript.**

We thank the reviewer for their suggestion of providing an explanation of the segmentation process. Trees were not segmented for this paper; we used data that had already been segmented by the authors for a separate study (Owen et al., 2021), and we apologise for the lack of clarity. We have amended the description of tree segmentation in section 2.5 (L218 – 221):

*"We used individual tree point clouds downsampled to 0.05 m, to aid computation time, and segmented using the automated tree segmentation program treeseg (Burt et al., 2019), implemented in C++,  by Owen et al., (2021) for that study.  Individual segmented tree data are available in Owen et al., (2022)."*

**2.18    L216 How to analyze the WAI after voxelizing woody point clouds? Some details should be introduced, which is key to calculating É' .**

We thank the reviewer for their comment and apologise for the lack of clarity in our methods for calculating individual tree WAI. Individual tree WAI was calculated in the same way (voxel classification method) as individual tree PAI, but using the wood-only cloud from the wood – leaf separation step. We have changed L232-233 in section 2.5:

*"In the same way as for PAI, we calculated WAI using the separated wood point cloud within the projected crown area of the whole tree (Figure 2d; using the whole crown and not just the wood point cloud)"*

**2.19    L225 Why explore the relationship between PAI and CAI in this study? The CAI assessment seems to deviate from the research topic, as it is not highly related to LAI and WAI but to the crown projection area, except the canopy gap area. Moreover, using images for CAI analysis is sufficient.**

We apologise for not clearly stating our justification for the use of CAI in this study. CAI is used in this study as an indicative measure of both stand density and local competition, and is included to both explore how PAI is affected by competition, but also to correct for anticipated competitive affects that would otherwise impact our conclusions on species' differences in alpha. We thank the reviewer for their comment, and have amended our manuscript section 2.6 (L245 – 248).

Please see also our response to comment 2.8.

In closed canopies or canopies with crown overlap imagery would not capture CAI, since CAI is calculated using the sum of all projected crown area, not only that visible from imagery. We used TLS to measure CAI as CAI estimates are generated from the sum of all tree crown projected area and so requires individual tree measurements, either from segmented TLS or from ground measurements.

**2.20    L245 As shown in Figure 3, PAI estimated using the LiDAR Pulse method more strongly agreed with DHP PAI than the Intensity Image method. However, I found their correlation ($R^2$) is not particularly significant.**

We believe this is a misreading of our meaning, and therefore apologise for not using clear language in reporting our statistical results. We have therefore amended section 3.1 (L266-268):

*"Of the two single scan TLS methods tested (LiDAR Pulse method and 2D Intensity Image method), we found that the relationship between PAI estimated using the LiDAR Pulse method and DHP PAI, had a higher $R^2$ than the 2D Intensity Image method"*

**2.21    L248 Please carefully recheck the description of the results is correct according to Figure 3. As shown in Figure 3a, the Pulse-based method overestimates the PAI, while the intensity-based method underestimates the PAI.**

We apologise for the lack of clarity in explaining these results and meant to say that both methods underestimate relative to DHP at larger values. We have therefore amended the sentence in section 3.1 (L270 -271):

*"At larger PAI values, relative to DHP, both TLS methods underestimated PAI compared with DHP (Figure 3b)."*

**2.22    L264 You did not label Voxel-Based PAI in Figure 3. Do you mean the TLS PAI**

We apologise for the lack of clarity in this sentence. Section 3.2, L286 is referring to whole plot plant area index (Figure 4), which does include Voxel-Based PAI. We have now removed the reference to Figure 3 from this sentence.

**2.23    L269 Maybe you did not set a suitable threshold when defining blank voxels. Merely my speculation!**

We thank the reviewer for their speculation on why we may be experiencing overestimation from Voxel-Based PAI estimates. We followed standard protocol as described in the published literature for the Voxel-Based, LiDAR Pulse and 2D Intensity Images methods, which has allowed us to draw a fair comparison between derived PAI values from each method. Although threshold values could be influencing PAI estimates, we have made a 'best choice' to classify non-zero point containing voxels as vegetation and believe that, while important research, further exploration of threshold values would be beyond the scope of this study.

Please see also response to comment 2.15.

**2.24    L282 and 257 You forget to mark the 1:1 dash line in these figures.**

This graph (Figure 4b) presents the variation in PAI against CAI in order to understand how competition and stand density affects PAI so we would not assume a 1:1 relationship. Dashed line on Figure 3b is at 0 to highlight systematic variation in the residuals.

We apologise for the lack of clarity and have amended L308 to make this clearer:

*"Dashed line in panel a represents 1:1 relationship"*

In Figure 3b, the dashed line represents 0, as this panel is showing the relationship between TLS residuals and DHP PAI. We apologise for the lack of clarity and have amended L280:

*"Dashed line in panel a represents 1:1 relationship."*

And L281:

*"dashed line in panel b represents 0"*

**2.25    Although authors used the published woody-and-foliage separation methods, it is necessary to display some examples of TLS separation results scanned from diverse plots grown with different species. Due to the lack of validation data, it may be challenging to evaluate the segmentation accuracy. However, presenting the separation results is still available to support visual evaluation.**

We agree that showing an example visual assessment of wood/ leaf separation is beneficial to the reader and have included an example of a wood/ leaf separated *P. sylvestris* in Figure 2, panels a and b. We have now included signposting in the figure caption, L212-213:

*"Panels a and b show wood and leaf separation of an example P. sylvestris, carried out using TLSeparation (Vicari et al., 2019)."*

We also note that wood – leaf separation was carried out by the authors for a separate published study (Owen et al., 2021). We apologise for the lack of clarity and have changed our description of the wood – leaf separation process accordingly in section 2.5 (L223-225). Please see our response to comment 2.33 below.

**2.26    It is not easy to accurately separate the branch and leaf point clouds of trees except those of broadleaf. More importantly, I am worried about whether it is applicable to use the same voxel size to calculate the WAI of different tree species, which is crucial to the conclusion.**

Although we agree it may theoretically be more difficult to separate wood and leaf in needleleaf trees, we note that TLSeparation was developed with applicability to both types of trees, with separation difficulties attributable to scanning strategy rather than separation algorithm (Vicari et al., 2019b). We note that this problem was minimised to the best of our ability in our dataset, as we followed a dense scanning strategy as outlined in Wilkes et al., (2017). As for comment 2.25 above, we have included an example visual assessment of a wood – leaf separated (needleleaf) *P. sylvestris* and included signposting in L212-213:

*"Panels a and b show wood and leaf separation of an example P. sylvestris, carried out using TLSeparation (Vicari et al., 2019)."*

**2.27    L294-297 These sentences are not clear. How to assess tree-specific drought tolerance? You would better add some description about its evaluation methods and list the metrics to evaluate the drought tolerance of different tree species in this figure and the related references.**

We agree the source of drought tolerance rankings needs to be clear and apologise for omitting this in the figure caption. Drought tolerance are taken from the widely-cited Niinemets and Valladares, (2006). We have amended L326:

*"Drought tolerance rankings are taken from Niinemets and Valladares, (2006)"*

**2.28    In section 4.1, why did you discuss the plot-scale CAI variation? The topic of this section is comparing diverse approaches to deriving PAI.**

We apologise for the lack of clarity around the role of CAI in this study. As CAI was used as an indicative measure of stand density and local competition, it was discussed in this section as plots with higher CAI (and therefore greater stem density) showed greater variation in estimated PAI values from each method/ sensor. We note that we have now updated our description of CAI and its role in this study in section 2.6 (L245 – 248) and hope that the discussion in section 4.1 is now more clear.

Please see also our response to comment 2.8.

**2.29    The title of Section 4.2 is a phenomenon that you need to analyze. Sections 4.2 and 4.3 still belong to Section 4.1 to discuss the LiDAR-extracted metrics with that of DHP.**

The titles of sections 4.2 and 4.3 were intended to emphasise findings of particular interest and relevant to the initial aims of this manuscript. We agree, however, with the reviewer that these sections belong with 4.1 and have removed these section titles to make this more clear.

**2.30    L320 According to the field data and Figure 3, what is a very low PAI value? Providing a quantitative indicator will significantly improve the manuscript's readability than using adjective words.**

We thank the reviewer for the comment and agree that more quantitative language would improve the manuscript. We have therefore amended L342 to say:

*"except at very low PAI values ($PAI_{TLS} < 0.5$)."*

**2.31    L348 The highest $R^2$ does not show a strong correlation.**

We thank the reviewer for their comment and agree that we have not used clear statistical language. We have therefore changed the wording in L377 – 378:

*"The relationship between the LiDAR Pulse method and TLS derived CAI had the highest $R^2$"*

**2.32 L374 This sentence is not clear. "Although species explain some variation in α, tree height and plot CAI were stronger predictors for all species…." According to the principle of these parameters, it is hard for me to agree that CAI and WAI have a strong correlation.**

We agree with the reviewer that we have not used clear statistical language. We have therefore changed this sentence in section 4.5 (L403):

*"Although species had a weak relationship with α, tree height and plot CAI had a statistically significant relationship with α (p<0.001 – p<0.05) for all species, showing the importance of local stand structure on leaf and woody allocation."*

**2.33 L390-392 This is an interesting point. I prefer you to provide some figures and statistical information to prove your finding, especially in different plots with variable growing patterns (growing density, CAI, and WAI related to the tree species, as you mentioned in the Conclusion section). It is beneficial to deepen this study topic.**

We thank the reviewer for finding this point interesting and their suggestion of including quantitative results of wood – leaf separation. The discussion point the reviewer refers to is a reference to the published paper describing the wood – leaf separation algorithm. Due to the lack of validation data, evaluating quantitatively the effectiveness of the wood – leaf separation algorithm over the different tree sizes/ growing conditions is not possible for this study.

We note that the wood leaf separation process was carried out by the authors, for a separate study (Owen et al., 2021), in which the results are discussed in more detail and segmented tree files made available online, cited as Owen et al., (2022). We apologise for the lack of clarity here and have reworded our description of the wood – leaf separation process in section 2.5 (L223-229):

*"To estimate PAI, WAI and α for each tree, we used individual tree point clouds wood – leaf separated by Owen et al., (2021) using the open source Python library TLSeparation (Vicari et al., 2019a), and then used the separated wood point clouds to calculate WAI. TLSeparation assigns points as either leaf or wood, iteratively looking at a predetermined number of nearest neighbours (knn). The knn of each iteration is directly dependent on point cloud density, since high density point clouds will require higher a knn (Vicari et al., 2019a). The utility package in TLSeparation was used to automatically detect the optimum knn for each tree point cloud."*

**2.34 L 398 I agree that correcting WAI can improve the LAI assessment. The TLS-extracted data can support calibrating LAI based on WAI and PAI. The WAI may be similar among single trees of the same tree species. According to your results, the WAI shows a more evident relationship to tree height and stand density. I think the assessed WAI and plot-level PAI can be used to correct regional LAI for the plot or large-scale forests that were growing with limited tree species.**

We agree with the reviewer that interspecific WAI values will be of interest to some readers and thank the reviewer for their suggestion, which we think has significantly improved our manuscript. We have included these additional analyses in Appendix C as also discussed in response to comments 2.4 and 2.10. We hope this work sparks further research on improving LAI estimates at large scale.

**Some text errors that needed to be corrected are listed as follows:**

Thank you for pointing out these errors.

- **Do not use an abbreviation in the title of your manuscript, as many readers in other fields do not know the meaning of TLS.**

We agree with the reviewer that abbreviations should not be used in titles and have amended our title accordingly.

- **I suggest authors unify the reference format throughout their manuscript. Different citation formats appear in the same paragraph may confuse readers.**

We thank the reviewer for their comment and have checked and corrected referencing throughout.

- **L135 What are FunDIV plots?**

Added "Functional Diversity"

- **L142 I do not understand "altitudinal gradient 840 – 1400 m.a.s.l.".**

Changed to "altitudinal range"

- **L167 compare –ï¼ž compared**

Changed to "compared"

- **L169 and 180 Please note the font size of the subscript in the Pgap. This abbreviation can also be used in line 162.**

Changed to subscript and moved abbreviation to first use.

- **L176 Please add a comma to this sentence.**

Comma added

- **L199 Where are the solid black voxels in Figure 2?**

Changed to "Coloured voxels (green represents leaf and brown represents wood) are filled voxels and grey lines are empty voxels."

- **L209 wood only point clouds?**

Change to "separated wood cloud"

- **L210 TLSeparation classifies points as leaf or wood? This sentence is not clear.**

Changed to "*TLSeparation* assigns points as either leaf or wood"

- **L219 TLS PAI and DHP PAI? (Using PAI$_{TLS}$ and PAI$_{DHP}$ instead)**

Changed throughout.

- **L234 Please add a comma to this sentence.**

Comma added.

- **L246-248, L264-265 You can mark these metrics in the insets of Figure 3.**

We think it is important to refer to statistical results in the main text of the manuscript for emphasis, however have included them in the figure captions as well for completeness.

- **Points in Figure 3 can be denoted as different marks or colors, such as circles or crosses, red or blue, to make this chart clearer (like the style of Figure 4).**

Changed to circles and triangles.

- **L262 Please unify the term throughout the manuscript. I think TLS whole plot PAI means TLS PAI(PAI$_{TLS}$).**

Changed to "whole plot PAI$_{TLS}$"

- **L274-276 You would better mark these metrics in the subfigures of Figure 4.**

We think it is important to refer to statistical results in the main text of the manuscript, however have included them in the figure captions as well for completeness.

- **In Figures 3 and 4, please delete the unit of PAI. The PAI, LAI and WAI are all ratio-type parameters (no need to denote unit).**

    Removed units and changed axis labels to new subscript (PAI$_{TLS}$ / PAI$_{DHP}$)

- **L318 TLS – DHP comparisons?**

Changed to "studies comparing PAI$_{TLS}$ with PAI$_{DHP}$"

- **In this article, authors used lots of open-source software to support their analysis. I suggest they list all applicable packages and download links to make readers easy to use these tools.**

We thank the reviewer for their suggestion of providing a summary of all open-source software used for this manuscript. We have cited all the software used in text and in the reference list at the end of the manuscript. We believe that citing packages in the main bod, readers are able to get a more detailed and contextualised explanation of the use in individual software packages.

- **Please carefully check the format of all references according to the manuscript preparation guidelines and the latest published papers in Biogeosciences. The current reference format needs to be optimized.**

We thank the reviewer for their comment and have checked the reference format.

2006); we chose SMA because we do not have a 'true' validation dataset, so avoid assuming either DHP or any
of the TLS methods produces the most accurate results. For each TLS method, we assessed the relationship with
DHP using the coefficient of determination and RMSE. We chose to compare PAI values rather than WAI or LAI
as each method corrects for non-photosynthetic elements in different ways and would introduce bias, limiting the

[revised manuscript text omitted]

**Appendix C**

$$WAI = m_{species} + b \quad (1)$$

$$WAI = m_{height} + b \quad (2)$$

$$WAI = m_{CAI} + b \quad (3)$$

$$WAI = m_{PAI} + b \quad (4)$$

Where WAI is the wood area index, *species, height, CAI* and PAI are the tree species, tree height, crown area index of the plot in which the tree is growing and tree plant area index respectively and *m* and *b* are parameters to be fit.

[Figure]

**Figure 2: Linear model derived WAI values (m, equation C1) for all 2472 individual trees of species *P. sylvestris, P. nigra, Q. faginea, Q. ilex* and *P. pinaster*. Error bars represent 95% confidence intervals. Species are listed from low – high drought tolerance, with the exception of *P. pinaster*, for which drought tolerance index has not been calculated in the literature.**

**Table 3: Linear model (equation C1) showing relationship between tree species and WAI for all 2471 individual trees. Significance codes: p < 0.001 '***'; p < 0.01 '**'; p < 0.05 '*'; not significant 'ns'**

| Species | *m* (eq. 1) | Std. Error | P value |
|---|---|---|---|
| *P.nigra* | 0.57 | 0.008 | *** |
| *P. pinaster* | 0.69 | 0.018 | |
| *P. sylvestris* | 0.56 | 0.014 | |
| *Q. faginea* | 0.39 | 0.010 | *** |
| *Q. ilex* | 0.37 | 0.013 | *** |

**Table 4: Linear models (equations C2, C3, C4) predicting WAI as a function of tree height, CAI (density) and PAI**
**Significance codes: p < 0.001 '\*\*\*'; p < 0.01 '\*\*'; p < 0.05 '\*'; not significant 'ns'**

|  | *m* (eq. 2, 3, 4) | R$^2$ | P value |
|---|---|---|---|
| Tree Height | 0.02 | 0.27 | \*\*\* |
| CAI | 0.39 | 0.78 | \*\*\* |
| PAI | 0.11 | 0.35 | \*\*\* |

---

## Author Response (AR1)

**1. Author's response to reviewer 1**

We thank the reviewer for their comments, which we have discussed in the responses below and which we believe have significantly improved the manuscript.

**1.1 Regarding the methods, I am afraid that the authors have overlooked a significant part of the recent scientific literature on the subject. The voxel-based approach which is presented, tested and discussed in the manuscript is not an undisputed reference method and has a number of known drawbacks.**

We agree with the reviewer's view that a "best approach" to voxel-based PAI estimation remains contested. However, the aim of this paper is not to evaluate all possible voxel methods, but rather use a method with broad applicability to multiple TLS configurations. We chose the voxel-based method used in this study for clear reasons. First, we wanted to use full plots and segmented trees, so methods developed with single scans were inappropriate. Further, many radiative transfer methods require information on scanner location and beam direction limiting use to single scans or individual trees with known scan locations around them – not available in many TLS datasets. In addition, our preference, where possible, is to use methods that have been thoroughly and independently validated – in this case the voxel method chosen has been validated with destructive sampling. Finally, we only use methods that were open source and easily reproducible, excluding many insufficiently documented, GUI-based or proprietary approaches. Further discussion of our choice is given in our answer to the following comment.

We note that efforts to move towards a best practice consensus are building within the community requiring a dedicated effort and we believe our study provides direction for ways forward. We also highlight that methods should be compared across sensor and forest types to draw robust conclusions. As the data used in this study are published, we would be delighted to see further exploration of this topic making use of different voxelisation schemes but see the testing of these different methods to be beyond the scope of this study.

**1.2 L103-106 the authors briefly mention that there are different approaches to voxel-based estimation of PAI/LAI and they opt for one that treats elementary voxels as either empty or full (opaque). Unfortunately, there is no obvious justification for such a choice being made. The emerging consensus in the recent literature seems to be in favour of what the authors refer to as "simulating radiative transfer within each cube". One significant advantage being that the laser scanning geometry is considered, and hence the variable sampling intensity and occlusion effects on PAI estimated can be accounted for.**

While we appreciate the reviewer's comment that significant recent progress has been made in the field, our view is that a consensus on best approach is yet to be reached and the approach proposed by the reviewer is still contested (and please see the proceeding comment). In particular, there is increasing recognition that voxel size significantly influences PAI/WAI/LAI estimates, and many methods do not provide clear guidance on how to deal with this. For example, You et al., (2022), published after the submission of this manuscript, argue that voxel-based methods are highly sensitive to voxel size and present a morphology-based method to obtain LAI from the surface area on envelope fitting to extracted leaf points. A key benefit of the voxel-based approach used in this study is the clear justification for matching voxel size to point cloud resolution, as evaluated in Li et al., (2016) and validated using destructive samples. Using a radiative transfer approach PAI estimates are highly unstable over varying voxel sizes and there is no clear guidance from the literature on how to choose the correct one. Evaluating the many potential methods for

calculating LAI from TLS data are well beyond the scope of this study, however, we hope that the work presented here will contribute towards a future consensus in the field.

As discussed in our response to reviewer 2 (comment 2.9), we have amended our discussion of voxel size in section 1.3 to reflect the debate around voxel size choice.

We have added to section 2.4 to clarify our justification for use of a voxel classification approach over a radiative transfer approach, also commented on by reviewer 1.

**1.3 In addition, degrading point cloud resolution down to the voxels resolution is likely to degrade the quality of point cloud segmentation into leaf and wood as well as the PAI estimates.**

Downsampling is a critical step in *treeseg* (Burt et al., 2019) to handle computational loads associated with segmenting point clouds. We thank the reviewer for drawing attention to our lack of clarity over the justification for down sampling data and have added statement to section 2.5.

The requirement of individual tree point clouds in *TLSeparation* means downsampled individual tree point clouds are necessary without upscaling the resolution of individually segmented trees. The scale of this study (2472 trees, 33 plots) means using a raw data resolution is computationally impracticable and consequently, downsampling is common practice in studies using large datasets of individual tree point clouds. We believe choosing a *knn* based on the point cloud resolution is a robust approach to optimising wood leaf separation under the constraints associated with large datasets. We explain the *knn* in section 2.5 of the proposed manuscript.

We chose point cloud resolution as a trade-off between computational demands while retaining the structural information contained in each tree. We then matched voxel size to point cloud resolution rather than down sampling point cloud resolution to desired voxel size. This is in line with recommendations for the method we chose; many voxel-based methods provide no clear guidance on this. We thank the reviewer for drawing attention to our lack of clarity here and have reworded the explanation in section 2.4.

**1.4 For these reasons, I believe the conclusions drawn are not well grounded. The general conclusion that "Our results demonstrate the challenges that stand in the way of large scale adoption of TLS for vegetation indices monitoring" which refers to the large discrepancies observed between methods in their study contradicts recent papers such as (Béland and Kobayashi, 2021; Nguyen et al., 2022). Obviously, there are still challenges to address but this study does not seem to identify the real caveats associated with the use of TLS in vegetation studies.**

From the cited literature we assume the reviewer is referring to (1) voxel size and (2) occlusion.

Regarding voxel size, we agree that there is a major problem in choosing voxel size with little consensus on how to choose the correct one for a range or forest types and ecosystems (see responses above). Separate analysis performed within our group shows unstable results over a range of voxel sizes using a radiative transfer approach, with a wide range of derived indices for one scan across relatively small variation in voxel size, suggesting high model sensitivity to this input parameter. The method we chose matches the voxel size to the resolution of the point cloud, and while the reviewer has pointed out there are "a number of known drawbacks" with this method, we feel that in the absence of well justified methods this is a pragmatic approach to accurately choosing the correct voxel size, and has been validated with destructively sampled data.

Regarding occlusion, Béland and Kobayashi, (2021) have chosen a very dense scanning density (5 m between scans), which is impractical in large-scale forest plots, and greater than the suggested scanning density in Wilkes et al., (2017), making such a dataset rarely available. Béland and Kobayashi, (2021) also suggest site specificity for their results, focusing on broadleaf trees, limiting the applicability of findings to our mixed Mediterranean forest.

Finally, the conclusion reached in our paper that "challenges stand in the way of large scale adoption of TLS" are drawn from a comparison of three TLS methods with conventional DHP. Neither papers cited (Béland and Kobayashi, 2021; Nguyen et al., 2022) test a voxel-based method against other widely used TLS PAI derivation methods (e.g. LiDAR pulse, 2D intensity image) and DHP, rather they are focused entirely on a voxel-based approach. We therefore argue the novelty of our findings and believe they do not contradict these papers.

**1.5 Both theory and algorithms have advanced significantly in recent years and convergent approaches to PAI/LAI estimates from lidar (both TLS and ALS) are emerging. Maybe the authors will want to check the following references**

We thank the reviewer for the references provided, however, argue that our dataset is significantly different from the data used in these studies. Methods suggested by the reviewer have been developed with individually scanned trees or branches (e.g. Béland et al., 2011; Soma et al., 2018), or with simulated data (e.g. Grau et al., 2017; Pimont et al., 2019, 2018; Soma et al., 2020). Individually scanned trees or branches can be scanned with a set of known scan positions allowing the precise location, distance, and beam angle from the scanner to be derived. Further, Béland and Kobayashi, (2021) focused on broadleaf trees functionally and physiologically different to those in our study, used a prohibitively dense scanning strategy (5 m), and lack validation from destructive sampling. Our dataset comprises 2472 trees scanned from 528 locations. To derive point-level information containing scanner location and beam angle would add significant complexity and computational load to the study. While an important question, understanding the necessity for this added complexity is beyond the scope of this paper.

As stated in the proposed manuscript L26-28: *"Our findings highlight the value of TLS data to improve fundamental understanding of tree form and function, but also the importance of rigorous testing of TLS data processing methods at a time when new approaches are being rapidly developed.",* we argue that the purpose of this paper is not to evaluate the latest methods, rather to take a step back and test existing methodologies with a large dataset.

**1.6 the authors refer to Beland et al. 2014 when noting the potential role of voxel size in the voxel-based approach, but that paper uses a voxel-based approach which is not the one used by the authors**

Thank you for pointing out this inappropriate reference to Béland et al., (2014); we apologise for this mistake and have corrected it, changing the reference to Li et al., (2016) who found voxel size to have significant effect on PAI estimates using the same voxel-based approach used in the study.

**1.7 Regarding the ecological insights, the clearest result seems to be that the alpha parameter (WAI/PAI) decreases with tree size (figure 6). The interpretation of what may appear as a paradox is largely speculative. It is interpreted as the result of competition but no data supporting this is presented. One might have tried to explore how alpha evolved in relation to the local competition index for instance.**

We thank the reviewer for the suggestion of exploring how alpha evolves with local competition, which is a key finding of this paper that we have not sufficiently highlighted. Figure 6b shows how alpha changes in relation to plot-level crown area index (CAI), a measure of the plot area covered by tree crown, and one that we have used as a proxy measure of local competition.

To clarify the use of CAI as indicative measure of local competition, we have changed the wording in section 2.6.

**1.8 There is abundant literature (and theoretical arguments) that indicate that LeafToWood biomass ratio of trees growing in stands will tend to decrease with size (Bartelink, 1997; Forrester et al., 2017; Mensah et al., 2016). In the present study, the WoodToLeaf area ratio is found to decrease with tree size (for the four species for which there is a significant trend in figure 6). This could be an artefact as the authors point out (l. 387-394). The issue might indeed have to do with the leaf/wood filtering.**

We agree with the reviewer that there is abundant literature that argue leaf to wood biomass ratio will tend to decrease with size, however, the literature cited by the reviewer differ fundamentally from our study in ways that may explain differences in results. For example, the focus species, *Fagus sylvatica* in Bartelink, (1997) is functionally different to species analysed in this study; Forrester et al., (2017) evaluate leaf biomass rather than wood to plant ratio and Mensah et al., (2016) omit correction for competition in their models while also excluding the largest trees from the study possibly introducing bias. This means that the arguments presented may not hold in our dataset measured in a mixed Mediterranean forest.

We agree that wood to plant ratio could be influenced by an artifact of wood – leaf classification, and have elaborated on this point in L418-425 of the proposed manuscript: *"Wood may be harder to accurately classify than leaves in TLS data (Vicari et al., 2019a), resulting in a higher occurrence of false positives in wood clouds, potentially leading to an overestimation in WAI, and therefore underestimation of α, especially in trees with small leaves which are prevalent in dry, Mediterranean environments (Peppe et al., 2011). The problem of misclassification will increase in taller trees due to TLS beam divergence, occlusion and larger beam footprint at further distances (Vicari et al., 2019a), suggesting that WAI overestimation could be more pronounced in tall trees. Although our dense scanning strategy (Owen et al., 2021) was designed to mitigate some of these effects, it is possible our findings could underestimate the slope of the negative relationship between α and tree height.".* Based on this, we would expect to be underestimating the negative slope of the relationship between alpha and tree height if it was an issue of misclassification.

**1.9 This is also a field where progress has been made in recent years and maybe the authors would want to test alternative algorithms to TLSeparation which might perform better on their data. Some pointers are given below**

We agree that there has been progress in the field of wood – leaf classification, however, we argue most progress has been focused on scaling wood – leaf classification from individual trees to whole scan or plot data (e.g. Krisanski et al., 2021; Wan et al., 2021; Wang, 2020; Wang et al., 2018; Wu et al., 2020) rather than major improvements in the classification framework itself. In the case of LeWoS (Wang et al., 2020), the tool has been tested only with tropical trees and, although, distributed as open-source, is either in the form of Matlab code or a pre-compiled executable, substantially limiting wider applicability. Testing the multitude of available approaches to wood – leaf classification would be invaluable to the

field, however, is beyond the scope of this study – not least because such a test should use destructively sampled validation data, which we do not have access to. Here we are interested in using well-established methodology that has been validated with a range of tree types, so based our choice on that criteria.

**1.10 My overall appreciation is that the data collected is very significant and could indeed contribute some new insights in terms of tree/forest ecology but more work is needed prior to publication.**

We thank the reviewer for their comments and appreciate that the reviewer recognises the significance of our data and results. We are confident that following their and the other reviewer's comments that the manuscript has been significantly enhanced.

**1.11 Reprocessing the TLS data already segmented using an open source freely available code incorporating much of the latest theoretical improvements should not take long. This analysis may profoundly alter the reported results (i.e the large overestimation of PAI with a voxel-based approach and the unexpected negative trend in WAI/PAI with increasing tree size). This may help clarify whether leaf/wood segmentation may be an issue and require further scrutiny or not.**

Whilst additional analyses are always possible, in this case we believe our methodological choices are defensible, and these have been discussed in previous responses. We use well-established and tested leaf separation and PAI estimation methods that were tested, in the case of voxel-based method, with destructive samples. The scope of this study is to benchmark the most rigorously available methods, not testing all available methods but taking the conservative approach. Further, all the methods tested in this study are either open source in common programming languages, or, where we have written code this has been made freely available. Not all the methods suggested by the reviewer are open source or easily integrated into automated workflows. We believe that running the analysis again would introduce new, different biases, and don't believe this would enhance manuscript without changing scope.

**2. Author's response to reviewer 2**

We thank reviewer for acknowledging the impact of this paper and their comments which we have discussed below and think have significantly improved the manuscript.

**2.1 I think this article's first and foremost improvement point is that the research goal is not clear and in-depth enough.**

We apologise for a lack of clarity here, and agree that the twin goals of methodology comparison and ecological insight are not presented in as clear a manner as they could be. We have edited the wording in section 1.3 to improve their readability and enhance the communication of their importance.

**2.2 The presentation of the results is not complete, which makes it difficult for readers to capture their needed information, such as the WAI variation trend (functions) among 5 tree species related to their height and density.**

We apologise that our analysis of WAI was not clear to the reviewer. We chose to compare methods based on PAI estimates, and not WAI or LAI, to avoid introducing additional processing steps and complexity and therefore to more directly compare the chosen methodological approaches. Differences in PAI between different TLS and DHP estimates can be attributed to differences in processing approaches, whereas comparison of WAI introduces additional error from separation approaches. To improve clarity, we have added a statement to section 2.6

See also our response to comment 2.4 below.

**2.3 Referring to the DHP results, authors evaluated the error of WAI and PAI analyzed by point clouds. I would like to know if the authors use the TLS data to improve the LAI evaluation accuracy. TLS can support assessing the single tree and plot-level WAI more accurately.**

We agree completely that the combination of TLS and DHP might improve analyses, and this has been developed in methods not tested in this paper (e.g. Kamoske et al., 2019).

Here we did not use the two datasets together, preferring instead to retain the ability to compare them as independent estimates of the indices of interest. We note that neither should be viewed as the 'truth', and therefore using them in combination could introduce additional biases that would be challenging to disentangle. Nevertheless, others could use our data to perform the analyses suggested.

**2.4 More importantly, whether the WAI of different Mediterranean trees has similarities between the same species, as well as providing specific information (maybe list in thematic tables to show the relationship among species, tree height, density, and PAI), will make readers benefit greatly. I think the measured data of this study can support this research goal, while they are not fully presented in the current edition.**

Although these are not the focus of this study, we agree that additional information could prove useful to some readers. We thank the reviewer for their suggestion of including WAI analysis, which we think has significantly improved the manuscript. We have added Figure C2 and Tables C3, C4 to the supplementary information and refer to this in the main manuscript, section 3.4.

**2.5 In addition, the presentation of the results is incomplete. I did not find the location, site conditions and tree species appearance of the measured plots shown in the manuscript.**

We apologise for this omission. This information is presented in the cited study Owen et al. (2021), but we have added a detailed site map to the supplementary materials, Figure 1, which is referred to in section 2.1 of the main text.

Please see also our response to comment 2.14

**2.6 The segmentation results of different tree species and the statistical information on PAI and WAI of trees grown in different site conditions were also not provided.**

We thank the reviewer for their comment and apologise for lack of clarity around the segmentation process. Individual tree segmentation was carried out by the authors for a separate study (Owen et al., 2021). We have amended section 2.5 to clarify the segmentation process and have signposted (Owen et al., 2021).

Please see also our response to comment 2.17.

**2.7 In addition, critical mathematical functions and quantitative conclusions are also lacking in the current edition.**

We apologise for this lack of completeness. We are not entirely clear to which functions the reviewer refers, but for reasons of clarity and brevity we chose to primarily describe the various processing methods we used rather than repeat their original descriptions, which are extensive within the cited literature. Where equations have been used from other studies, we have cited the original equation number along with the paper in-text (but see response to 2.8 below).

**2.8 I suggest authors reconsider whether it is necessary to study the CAI. This parameter can be easily analyzed using remote sensing images without using TLS.**

In this study we used CAI as a proxy measure of stand density, which was a necessary within our model to both understand and correct for the effect of stand density on wood to plant ratio, α. Controlling for stand density (using CAI as a proxy) is important as trees growing in dense plots have lower water availability per tree (see section 1). We chose to use CAI as Owen et al., (2021) showed that the metric is also indicative of plot-level competition and the metric accounts for crown overlap which cannot be estimated from imagery in closed forests. Furthermore, Coomes et al., (2012) showed CAI to be better than traditional metrics such as basal area, as it is more intuitive to non-specialists and strongly predicts productivity.

We apologise for the lack of clarity when describing this metric and its intended use in the study, which was also commented on by reviewer 1 (comment 1.7).

We have therefore amended our description of the key metric, CAI, for quantifying stand density and local competition in section 2.6.

**2.9 Furthermore, is it applicable to use a fixed voxel size when analyzing WAI? After all, different tree species have various canopy shapes and branch structure features. Adaptive adjusting the voxel size according to the point cloud density and the branch distribution trend may be more reasonable.**

We thank the reviewer for their comment on voxel size and agree that finding an appropriate voxel size a complex problem (discussed extensively in the response to the other reviewer). We chose the method of voxel classification rather than a radiative transfer approach as it has a definitive method for choosing voxel size based on matching the voxel size to the resolution of the point cloud, which was tested against voxel sizes based on individual tree leaf size, and distance of beam, using destructive samples in Li et al., (2016). Using a radiative transfer approach, the methodology for choosing the "correct" voxel size is not clear, and others' work (and our own additional, unpublished analyses) has shown that estimated PAI values are highly sensitive to voxel size choice.

We have amended our discussion of voxel size in section 1.2 to reflect the contentious debate around voxel size choice.

To clarify our justification for use of a voxel classification approach over a radiative transfer approach, also commented on by reviewer 1, we have added to section 2.4.

**2.10   Optimizing the TLS-based WAI assessment methods, summarizing the regulation of interspecific WAI variation, and using these rules to improve the LAI assessment will make this article more attractive to better support research in related fields.**

We thank the reviewer for their suggestion of including analysis of interspecific WAI variation, which we think is a valuable addition to the paper, and refer to our response to previous comments (2.2, 2.4, 2.11), where we have included these new analyses.

Here, we've focussed on interspecific variation in alpha and PAI, rather than WAI and LAI, but recognise that there would be value in such an additional set of analyses. We agree that developing new methods to correct for WAI in LAI estimates using approaches assessed in this paper would make for exciting work, however we think that to do this well we would require further testing and validation, ideally using destructive samples or multitemporal leaf on/leaf off remote sensing data, which is beyond the scope of this paper.

**The following are some detailed points. I hope they will help improve the current edition.**

**2.11   I suggest authors clarify their research goal in the initial section of the manuscript. As a reader, I am more interested in how to use TLS to analyze WAI. However, authors did not briefly introduce the WAI extraction methods in the abstract but focused on comparing point cloud extraction methods of PAI and LAI.**

We apologise for the lack of clarity in explaining our research goals. As in our response to comment 2.1, we have restated our primary and secondary research goals in section 1.3.

We thank the reviewer for their suggestion of analysing WAI, which we think has significantly improved the manuscript. As for comment 2.4, we have now included this analysis. We have chosen to keep the focus on comparing alpha, as this value is widely discussed in the literature. We have added a statement to this effect to section 2.5.

**2.12    They focused on the wood to total plant area (α). I wonder if it is feasible to measure the plant area because of the occlusion effect during scanning. TLS may be more suitable for analyzing WAI.**

We apologise not clearly stating the reasons for comparing PAI rather than WAI. All remote sensing methods evaluated in this paper (three TLS methods and DHP) more directly measure PAI than WAI or LAI as sensors are measuring the whole plant. Correcting for wood/ leaf to derive WAI/ LAI requires additional processing steps, which vary according to sensor (wood/ leaf separation algorithms for TLS and image masking for DHP, as these systems are not deciduous, and therefore leaf-off scans can't be made), introducing bias and limiting our ability to compare output. We have added a statement to this effect to section 2.6.

Please see also our response to comment 2.2

We agree that occlusion is a known problem with TLS data in closed canopy forests, however we have minimised the potential occlusion effects by following a dense scanning strategy following the widely cited Wilkes et al., (2017).

**2.13    Section 1.3 It will be more interesting to add some research topics on integrating the fine-scale WAI (or α) assessed based on TLS to correct the large-scale LAI extracted from the multi-source remote sensing images. Based on the high-quality field dataset, it should be feasible to use this research in optimizing the large-scale LAI distribution evaluation.**

We agree this is an exciting idea and could be the focus of follow-on work. We think that the work presented and, as the reviewer points out, our dataset provides a foundation for a more robust comparison of LAI and new insights from multi-source RS datasets, but that this would be an additional methodological development beyond the scope of our current study.

Following your suggestion, we have added a comment to this effect to section 4.4 of the revised manuscript.

**2.14    In Sections 2.1 and 2.2, the location map of study plots and some images showing the scene of plots should be provided. The pictures of tree species also need to be added to show their phenotypic characteristics, which is beneficial to evaluate their drought tolerance (L323-324).**

We agree with the reviewer that a location map of the study plots would be beneficial to the manuscript and thank them for the suggestion. We have therefore added a new figure, A1 to Appendix A showing the locations of plots within the two field sites, Alto Tajo and Cuellar in central Spain.

We believe that the plots used in this study are well studied and documented in the literature and therefore a detailed description of individual plot characteristics would repeat information already available. We have added signposting to this is section 2.1.

The five focus species of this manuscript are widely studied and known species and therefore believe that adding individual images of each species is unnecessary.

**2.15    L 191 When setting this threshold (> 0 points) to identify the filled voxels, did you filter noisy points out from the tree TLS datasets? It is not easy to identify and**

**filter all noise in TLS data. I am worried the noise would lead to a lower P$_{gap}$ and cause inaccurate LAI and PAI.**

We apologise for not making explicit the noise filtering process of our data. We denoised individual-tree point clouds using height dependant statistical filtering as outlined in Owen et al., (2021), and combined individual tree point clouds into whole plots. We have added a statement to this effect to section 2.4.

While any remaining noise may indeed lead to lower *P$_{gap}$*, we followed standard processing procedure for this voxel classification method outlined in Hosoi and Omasa, (2006) and tested using destructive samples in Li et al., (2016). Similarly, we followed standard protocol in the published literature for the other two methods (LiDAR Pulse and 2D intensity Image), and therefore consider that our work is a fair representation of each methods' ability to accurately derive PAI and allows a comparison of each methods' merits and drawbacks.

**2.16 L203-204 Some structure features of woody and foliage materials can be analyzed based on the pointset-, height bin-, and patch-based models. Please revise this sentence.**

We apologise for the lack of clarity in this statement, and thank the reviewer for their suggestion. What we meant to say was that the voxel-based approach was the only method compared in this study capable of analysing PAI, WAI and LAI of segmented individual tree point clouds. We have reworded the sentience in section 2.5 to make this clear.

**2.17 L206 The principle of TLS segmentation methods needs to be briefly introduced before the voxelization step. It is beneficial to improve the readability of the manuscript.**

We thank the reviewer for their suggestion of providing an explanation of the segmentation process. Trees were not segmented for this paper; we used data that had already been segmented by the authors for a separate study (Owen et al., 2021), and we apologise for the lack of clarity. We have amended the description of tree segmentation in section 2.5.

**2.18 L216 How to analyze the WAI after voxelizing woody point clouds? Some details should be introduced, which is key to calculating É' .**

We thank the reviewer for their comment and apologise for the lack of clarity in our methods for calculating individual tree WAI. Individual tree WAI was calculated in the same way (voxel classification method) as individual tree PAI, but using the wood-only cloud from the wood – leaf separation step. We have updated section 2.5.

**2.19 L225 Why explore the relationship between PAI and CAI in this study? The CAI assessment seems to deviate from the research topic, as it is not highly related to LAI and WAI but to the crown projection area, except the canopy gap area. Moreover, using images for CAI analysis is sufficient.**

We apologise for not clearly stating our justification for the use of CAI in this study. CAI is used in this study as an indicative measure of both stand density and local competition, and is included to both explore how PAI is affected by competition, but also to correct for anticipated competitive affects that would otherwise impact our conclusions on species' differences in alpha. We thank the reviewer for their comment, and have amended our manuscript section 2.6.

Please see also our response to comment 2.8.

In closed canopies or canopies with crown overlap imagery would not capture CAI, since CAI is calculated using the sum of all projected crown area, not only that visible from imagery. We used TLS to measure CAI as CAI estimates are generated from the sum of all tree crown projected area and so requires individual tree measurements, either from segmented TLS or from ground measurements.

**2.20  L245 As shown in Figure 3, PAI estimated using the LiDAR Pulse method more strongly agreed with DHP PAI than the Intensity Image method. However, I found their correlation ($R^2$) is not particularly significant.**

We believe this is a misreading of our meaning, and therefore apologise for not using clear language in reporting our statistical results. We have therefore amended our description of results in section 3.1.

**2.21  L248 Please carefully recheck the description of the results is correct according to Figure 3. As shown in Figure 3a, the Pulse-based method overestimates the PAI, while the intensity-based method underestimates the PAI.**

We apologise for the lack of clarity in explaining these results and meant to say that both methods underestimate relative to DHP at larger values. We have therefore amended the sentence in section 3.1.

**2.22  L264 You did not label Voxel-Based PAI in Figure 3. Do you mean the TLS PAI**

We apologise for the lack of clarity in this sentence. Section 3.2, this line is referring to whole plot plant area index (Figure 4), which does include Voxel-Based PAI. We have now removed the reference to Figure 3 from this sentence.

**2.23  L269 Maybe you did not set a suitable threshold when defining blank voxels. Merely my speculation!**

We thank the reviewer for their speculation on why we may be experiencing overestimation from Voxel-Based PAI estimates. We followed standard protocol as described in the published literature for the Voxel-Based, LiDAR Pulse and 2D Intensity Images methods, which has allowed us to draw a fair comparison between derived PAI values from each method. Although threshold values could be influencing PAI estimates, we have made a 'best choice' to classify non-zero point containing voxels as vegetation and believe that, while important research, further exploration of threshold values would be beyond the scope of this study.

Please see also response to comment 2.15.

**2.24  L282 and 257 You forget to mark the 1:1 dash line in these figures.**

This graph (Figure 4b) presents the variation in PAI against CAI in order to understand how competition and stand density affects PAI so we would not assume a 1:1 relationship. We apologise for the lack of clarity and have amended the caption of Figure 4 to make this clearer.

In Figure 3b, the dashed line represents 0, as this panel is showing the relationship between TLS residuals and DHP PAI. We apologise for the lack of clarity and have amended the figure caption.

**2.25    Although authors used the published woody-and-foliage separation methods, it is necessary to display some examples of TLS separation results scanned from diverse plots grown with different species. Due to the lack of validation data, it may be challenging to evaluate the segmentation accuracy. However, presenting the separation results is still available to support visual evaluation.**

We agree that showing an example visual assessment of wood/ leaf separation is beneficial to the reader and have included an example of a wood/ leaf separated *P. sylvestris* in Figure 2, panels a and b. We have now included signposting in the caption in Figure 2.

We also note that wood – leaf separation was carried out by the authors for a separate published study (Owen et al., 2021). We apologise for the lack of clarity and have changed our description of the wood – leaf separation process accordingly in section 2.5. Please see our response to comment 2.33 below.

**2.26    It is not easy to accurately separate the branch and leaf point clouds of trees except those of broadleaf. More importantly, I am worried about whether it is applicable to use the same voxel size to calculate the WAI of different tree species, which is crucial to the conclusion.**

Although we agree it may theoretically be more difficult to separate wood and leaf in needleleaf trees, we note that *TLSeparation* was developed with applicability to both types of trees, with separation difficulties attributable to scanning strategy rather than separation algorithm (Vicari et al., 2019b). We note that this problem was minimised to the best of our ability in our dataset, as we followed a dense scanning strategy as outlined in Wilkes et al., (2017). As for comment 2.25 above, we have included an example visual assessment of a wood – leaf separated (needleleaf) *P. sylvestris* and included signposting in the caption in Figure 2.

**2.27    L294-297 These sentences are not clear. How to assess tree-specific drought tolerance? You would better add some description about its evaluation methods and list the metrics to evaluate the drought tolerance of different tree species in this figure and the related references.**

We agree the source of drought tolerance rankings needs to be clear and apologise for omitting this in the figure caption. Drought tolerance are taken from the widely-cited Niinemets and Valladares, (2006). We have amended the caption in Figure 5.

**2.28    In section 4.1, why did you discuss the plot-scale CAI variation? The topic of this section is comparing diverse approaches to deriving PAI.**

We apologise for the lack of clarity around the role of CAI in this study. As CAI was used as an indicative measure of stand density and local competition, it was discussed in this section as plots with higher CAI (and therefore greater stem density) showed greater variation in estimated PAI values from each method/ sensor. We note that we have now updated our description of CAI and its role in this study in section 2.6 and hope that the discussion in section 4.1 is now more clear.

Please see also our response to comment 2.8.

**2.29   The title of Section 4.2 is a phenomenon that you need to analyze. Sections 4.2 and 4.3 still belong to Section 4.1 to discuss the LiDAR-extracted metrics with that of DHP.**

The titles of sections 4.2 and 4.3 were intended to emphasise findings of particular interest and relevance to the initial aims of this manuscript. We agree, however, with the reviewer that these sections belong with 4.1 and have removed these section titles to make this more clear.

**2.30   L320 According to the field data and Figure 3, what is a very low PAI value? Providing a quantitative indicator will significantly improve the manuscript's readability than using adjective words.**

We thank the reviewer for the comment and agree that more quantitative language would improve the manuscript. We have therefore amended this line.

**2.31   L348 The highest $R^2$ does not show a strong correlation.**

We thank the reviewer for their comment and agree that we have not used clear statistical language. We have therefore changed the wording.

**2.32   L374 This sentence is not clear. "Although species explain some variation in α, tree height and plot CAI were stronger predictors for all species…." According to the principle of these parameters, it is hard for me to agree that CAI and WAI have a strong correlation.**

We agree with the reviewer that we have not used clear statistical language. We have therefore changed this sentence in section 4.3 of the revised manuscript.

**2.33   L390-392 This is an interesting point. I prefer you to provide some figures and statistical information to prove your finding, especially in different plots with variable growing patterns (growing density, CAI, and WAI related to the tree species, as you mentioned in the Conclusion section). It is beneficial to deepen this study topic.**

We thank the reviewer for finding this point interesting and their suggestion of including quantitative results of wood – leaf separation. The discussion point the reviewer refers to is a reference to the published paper describing the wood – leaf separation algorithm. Due to the lack of validation data, evaluating quantitatively the effectiveness of the wood – leaf separation algorithm over the different tree sizes/ growing conditions is not possible for this study.

We note that the wood leaf separation process was carried out by the authors, for a separate study (Owen et al., 2021), in which the results are discussed in more detail and segmented tree files made available online, cited as Owen et al., (2022). We apologise for the lack of clarity here and have reworded our description of the wood – leaf separation process in section 2.5.

**2.34   L 398 I agree that correcting WAI can improve the LAI assessment. The TLS-extracted data can support calibrating LAI based on WAI and PAI. The WAI may be similar among single trees of the same tree species. According to your results, the WAI shows a more evident relationship to tree height and stand density. I think the**

**assessed WAI and plot-level PAI can be used to correct regional LAI for the plot or large-scale forests that were growing with limited tree species.**

We agree with the reviewer that interspecific WAI values will be of interest to some readers and thank the reviewer for their suggestion, which we think has significantly improved our manuscript. We have included these additional analyses in Appendix C as also discussed in response to comments 2.4 and 2.10. We hope this work sparks further research on improving LAI estimates at large scale.

**Some text errors that needed to be corrected are listed as follows:**

Thank you for pointing out these errors.

- **Do not use an abbreviation in the title of your manuscript, as many readers in other fields do not know the meaning of TLS.**

We agree with the reviewer that abbreviations should not be used in titles and have amended our title accordingly.

- **I suggest authors unify the reference format throughout their manuscript. Different citation formats appear in the same paragraph may confuse readers.**

We thank the reviewer for their comment and have checked and corrected referencing throughout.

- **L135 What are FunDIV plots?**

Added "Functional Diversity"

- **L142 I do not understand "altitudinal gradient 840 – 1400 m.a.s.l.".**

Changed to "altitudinal range"

- **L167 compare –ï¼ž compared**

Changed to "compared"

- **L169 and 180 Please note the font size of the subscript in the Pgap. This abbreviation can also be used in line 162.**

Changed to subscript and moved abbreviation to first use.

- **L176 Please add a comma to this sentence.**

Comma added

- **L199 Where are the solid black voxels in Figure 2?**

Changed to "Coloured voxels (green represents leaf and brown represents wood) are filled voxels and grey lines are empty voxels."

- **L209 wood only point clouds?**

Change to "separated wood cloud"

- **L210 TLSeparation classifies points as leaf or wood? This sentence is not clear.**

Changed to "*TLSeparation* assigns points as either leaf or wood"

- **L219 TLS PAI and DHP PAI? (Using PAI$_{TLS}$ and PAI$_{DHP}$ instead)**

Changed throughout.

- **L234 Please add a comma to this sentence.**

Comma added.

- **L246-248, L264-265 You can mark these metrics in the insets of Figure 3.**

We think it is important to refer to statistical results in the main text of the manuscript for emphasis, however have included them in the figure captions as well for completeness.

- **Points in Figure 3 can be denoted as different marks or colors, such as circles or crosses, red or blue, to make this chart clearer (like the style of Figure 4).**

Changed to circles and triangles.

- **L262 Please unify the term throughout the manuscript. I think TLS whole plot PAI means TLS PAI(PAI$_{TLS}$).**

Changed to "whole plot PAI$_{TLS}$"

- **L274-276 You would better mark these metrics in the subfigures of Figure 4.**

We think it is important to refer to statistical results in the main text of the manuscript, however have included them in the figure captions as well for completeness.

- **In Figures 3 and 4, please delete the unit of PAI. The PAI, LAI and WAI are all ratio-type parameters (no need to denote unit).**

  Removed units and changed axis labels to new subscript (PAI$_{TLS}$ / PAI$_{DHP}$)

- **L318 TLS – DHP comparisons?**

Changed to "studies comparing PAI$_{TLS}$ with PAI$_{DHP}$"

- **In this article, authors used lots of open-source software to support their analysis. I suggest they list all applicable packages and download links to make readers easy to use these tools.**

We thank the reviewer for their suggestion of providing a summary of all open-source software used for this manuscript. We have cited all the software used in text and in the reference list at the end of the manuscript. We believe that citing packages in the main text, readers are able to get a more detailed and contextualised explanation of the use in individual software packages.

- **Please carefully check the format of all references according to the manuscript preparation guidelines and the latest published papers in Biogeosciences. The current reference format needs to be optimized.**

We thank the reviewer for their comment and have checked the references throughout.

**3. References**

Bartelink, H.: Allometric relationships for biomass and leaf area of beech (Fagus sylvatica L), Ann. For. Sci., 54, 39–50, https://doi.org/10.1051/forest:19970104, 1997.

Béland, M. and Kobayashi, H.: Mapping forest leaf area density from multiview terrestrial lidar, Methods in Ecology and Evolution, 12, 619–633, https://doi.org/10.1111/2041-210X.13550, 2021.

Béland, M., Widlowski, J.-L., Fournier, R. A., Côté, J.-F., and Verstraete, M. M.: Estimating leaf area distribution in savanna trees from terrestrial LiDAR measurements, Agricultural and Forest Meteorology, 151, 1252–1266, https://doi.org/10.1016/j.agrformet.2011.05.004, 2011.

Béland, M., Baldocchi, D. D., Widlowski, J.-L., Fournier, R. A., and Verstraete, M. M.: On seeing the wood from the leaves and the role of voxel size in determining leaf area distribution of forests with terrestrial LiDAR, Agr. Forest Meterol., 184, 82–97, https://doi.org/10.1016/j.agrformet.2013.09.005, 2014.

Burt, A., Disney, M., and Calders, K.: Extracting individual trees from lidar point clouds using treeseg, Methods Ecol. Evol., 10, 438–445, https://doi.org/10.1111/2041-210X.13121, 2019.

Coomes, D. A., Holdaway, R. J., Kobe, R. K., Lines, E. R., and Allen, R. B.: A general integrative framework for modelling woody biomass production and carbon sequestration rates in forests, Journal of Ecology, 100, 42–64, https://doi.org/10.1111/j.1365-2745.2011.01920.x, 2012.

Forrester, D. I., Tachauer, I. H. H., Annighoefer, P., Barbeito, I., Pretzsch, H., Ruiz-Peinado, R., Stark, H., Vacchiano, G., Zlatanov, T., Chakraborty, T., Saha, S., and Sileshi, G. W.: Generalized biomass and leaf area allometric equations for European tree species incorporating stand structure, tree age and climate, Forest Ecology and Management, 396, 160–175, https://doi.org/10.1016/j.foreco.2017.04.011, 2017.

Grau, E., Durrieu, S., Fournier, R., Gastellu-Etchegorry, J.-P., and Yin, T.: Estimation of 3D vegetation density with Terrestrial Laser Scanning data using voxels. A sensitivity analysis of influencing parameters, Remote Sensing of Environment, 191, 373–388, https://doi.org/10.1016/j.rse.2017.01.032, 2017.

Hosoi, F. and Omasa, K.: Voxel-Based 3-D Modeling of Individual Trees for Estimating Leaf Area Density Using High-Resolution Portable Scanning Lidar, IEE T. Geosci. Remote, 44, 3610–3618, https://doi.org/10.1109/TGRS.2006.881743, 2006.

Kamoske, A. G., Dahlin, K. M., Stark, S. C., and Serbin, S. P.: Leaf area density from airborne LiDAR: Comparing sensors and resolutions in a temperate broadleaf forest ecosystem, Forest Ecol. Manag., 433, 364–375, https://doi.org/10.1016/j.foreco.2018.11.017, 2019.

Krisanski, S., Taskhiri, M. S., Gonzalez Aracil, S., Herries, D., and Turner, P.: Sensor Agnostic Semantic Segmentation of Structurally Diverse and Complex Forest Point Clouds Using Deep Learning, Remote Sensing, 13, 1413, https://doi.org/10.3390/rs13081413, 2021.

Li, Y., Guo, Q., Tao, S., Zheng, G., Zhao, K., Xue, B., and Su, Y.: Derivation, Validation, and Sensitivity Analysis of Terrestrial Laser Scanning-Based Leaf Area Index, Can. J. Remote Sens., 42, 719–729, https://doi.org/10.1080/07038992.2016.1220829, 2016.

Mensah, S., Glèlè Kakaï, R., and Seifert, T.: Patterns of biomass allocation between foliage and woody structure: the effects of tree size and specific functional traits, Ann. For. Res., 59, https://doi.org/10.15287/afr.2016.458, 2016.

Nguyen, V.-T., Fournier, R. A., Côté, J.-F., and Pimont, F.: Estimation of vertical plant area density from single return terrestrial laser scanning point clouds acquired in forest environments, Remote Sensing of Environment, 279, 113115, https://doi.org/10.1016/j.rse.2022.113115, 2022.

Owen, H. J. F., Flynn, W. R. M., and Lines, E. R.: Competitive drivers of inter-specific deviations of crown morphology from theoretical predictions measured with Terrestrial Laser Scanning, J. Ecol., 109, 2612–2628, https://doi.org/10.1111/1365-2745.13670, 2021.

Peppe, D. J., Royer, D. L., Cariglino, B., Oliver, S. Y., Newman, S., Leight, E., Enikolopov, G., Fernandez-Burgos, M., Herrera, F., Adams, J. M., Correa, E., Currano, E. D., Erickson, J. M., Hinojosa, L. F., Hoganson, J. W., Iglesias, A., Jaramillo, C. A., Johnson, K. R., Jordan, G. J., Kraft, N. J. B., Lovelock, E. C., Lusk, C. H., Niinemets, Ü., Peñuelas, J., Rapson, G., Wing, S. L., and Wright, I. J.: Sensitivity of leaf size and shape to climate: global patterns and paleoclimatic applications, New Phytol., 190, 724–739, https://doi.org/10.1111/j.1469-8137.2010.03615.x, 2011.

Pimont, F., Allard, D., Soma, M., and Dupuy, J.-L.: Estimators and confidence intervals for plant area density at voxel scale with T-LiDAR, Remote Sensing of Environment, 215, 343–370, https://doi.org/10.1016/j.rse.2018.06.024, 2018.

Pimont, F., Soma, M., and Dupuy, J.-L.: Accounting for Wood, Foliage Properties, and Laser Effective Footprint in Estimations of Leaf Area Density from Multiview-LiDAR Data, Remote Sensing, 11, 1580, https://doi.org/10.3390/rs11131580, 2019.

Soma, M., Pimont, F., Durrieu, S., and Dupuy, J.-L.: Enhanced Measurements of Leaf Area Density with T-LiDAR: Evaluating and Calibrating the Effects of Vegetation Heterogeneity and Scanner Properties, Remote Sensing, 10, 1580, https://doi.org/10.3390/rs10101580, 2018.

Soma, M., Pimont, F., Allard, D., Fournier, R., and Dupuy, J.-L.: Mitigating occlusion effects in Leaf Area Density estimates from Terrestrial LiDAR through a specific kriging method, Remote Sensing of Environment, 245, 111836, https://doi.org/10.1016/j.rse.2020.111836, 2020.

Vicari, M. B., Disney, M., Wilkes, P., Burt, A., Calders, K., and Woodgate, W.: Leaf and wood classification framework for terrestrial LiDAR point clouds, Methods Ecol. Evol., 10, 680–694, https://doi.org/10.1111/2041-210X.13144, 2019a.

Vicari, M. B., Pisek, J., and Disney, M.: New estimates of leaf angle distribution from terrestrial LiDAR: Comparison with measured and modelled estimates from nine broadleaf tree species, Agricultural and Forest Meteorology, 264, 322–333, https://doi.org/10.1016/j.agrformet.2018.10.021, 2019b.

Wan, P., Shao, J., Jin, S., Wang, T., Yang, S., Yan, G., and Zhang, W.: A novel and efficient method for wood–leaf separation from terrestrial laser scanning point clouds at the forest

plot level, Methods in Ecology and Evolution, 12, 2473–2486, https://doi.org/10.1111/2041-210X.13715, 2021.

Wang, D.: Unsupervised semantic and instance segmentation of forest point clouds, ISPRS Journal of Photogrammetry and Remote Sensing, 165, 86–97, https://doi.org/10.1016/j.isprsjprs.2020.04.020, 2020.

Wang, D., Brunner, J., Ma, Z., Lu, H., Hollaus, M., Pang, Y., and Pfeifer, N.: Separating Tree Photosynthetic and Non-Photosynthetic Components from Point Cloud Data Using Dynamic Segment Merging, Forests, 9, 252, https://doi.org/10.3390/f9050252, 2018.

Wang, D., Momo Takoudjou, S., and Casella, E.: LeWoS: A universal leaf-wood classification method to facilitate the 3D modelling of large tropical trees using terrestrial LiDAR, Methods Ecol Evol, 11, 376–389, https://doi.org/10.1111/2041-210X.13342, 2020.

Wilkes, P., Lau, A., Disney, M., Calders, K., Burt, A., Gonzalez de Tanago, J., Bartholomeus, H., Brede, B., and Herold, M.: Data acquisition considerations for Terrestrial Laser Scanning of forest plots, Remote Sens. Environ., 196, 140–153, https://doi.org/10.1016/j.rse.2017.04.030, 2017.

Wu, B., Zheng, G., and Chen, Y.: An Improved Convolution Neural Network-Based Model for Classifying Foliage and Woody Components from Terrestrial Laser Scanning Data, Remote Sensing, 12, 1010, https://doi.org/10.3390/rs12061010, 2020.

You, H., Li, S., Ma, L., and Wang, D.: Leaf Area Index Retrieval for Broadleaf Trees by Envelope Fitting Method Using Terrestrial Laser Scanning Data, IEEE Geoscience and Remote Sensing Letters, 19, 1–5, https://doi.org/10.1109/LGRS.2022.3214427, 2022.

---

## Referee Report (RR1)

I believe that the authors have made significant improvements to the manuscript. They have provided a more precise explanation of the study's objective in Section 1.3, and they have offered a more comprehensive review of previous research, including the impact of voxel size on vegetation indices calculations. In the Methods section, they have described the process of selecting the appropriate voxel size and referenced the drought tolerance rankings. Additionally, they have clarified the relationship between PAI and CAI and conducted further analysis to evaluate the relationship between WAI, species, height, CAI, and PAI. Their findings highlight the importance of accounting for variations in species mix and structure when correcting ground-based LAI estimates. However, there are a few minor improvements that they should consider:

- Please unify Beer-Lambert's law (Line 99) and Beer-Lambert law (Line 52).
- Unifying the reference format through the article would be better. For example: Li et al., (2016) to (Li et al., 2016).
- Please unify the unit used in the manuscript, such as 5 cm and 0.05 m.
- L262 You would better change the variable $a_s$ to $\varphi_s$, which makes readers distinguish it from $\alpha$.
- Please note to add commas or full stops after equations.
- Please recheck that the dashed line shown in Figure 3 is correct.
- Line 305 TLS-estimated. In Line 253, you defined CAI, so you can now use the abbreviation to refer to it more efficiently.
- Line 333 Please add a comma before "and PAI".
- Line347 Delete (CAI)
- Line 364 Please reconsider and declare this sentence:

  …trees in drier climates tend to have smaller leaves (Peppe et al., 2011), leading to more small canopy gaps…
- In my opinion, the voxel method is a more efficient way to extract local canopy structure features than searching for neighboring point clusters. However, this method may not capture all of the intricate details of the canopy structure.
- Line 423 Change to "the negative relationships between height and $\alpha$" and "the positive relationships between CAI and $\alpha$".
- Please ensure that the number of authors listed in each reference is consistent.
- Please label the values of b that appear in C2 to C4 in Appendix C. Also, remove the checkmarks that are used in these three equations.
- In Tables B1, B2, C3 and C4, what are 95% CI and ICC?

---

## Author Response (AR2)

**Authors' response to reviewer comments**

Referee 1

We wish to thank this reviewer for the time that they have taken to provide detailed comments on two versions of our manuscript through this revision process. We regret that the reviewer does not agree that the methods we have chosen to evaluate here have value, and that they are not willing to entertain the justification we have given either in the manuscript or in our previous responses. We do not agree with their conclusion that this invalidates our work, but we certainly welcome discussion of the challenges of developing robust and reliable processing methods for a discipline where working with multiple sensors and different ecosystems substantially impacts data structure. Robust methodological comparison and clarification of the validity of different methods under different conditions will only accelerate developments in the field. We believe our manuscript represents an important contribution to that discussion, and we are grateful that the other reviewers and editors see this, and that they do not agree with this reviewer's very negative response to our work.

We wish to reiterate that we are not attempting here to either evaluate all possible methods for TLS-based PAI estimation, nor are we attempting to definitively state which method is 'best' - with a lack of destructive ground truth information we believe that this would be foolhardy. Rather, we hope our manuscript provides those using these techniques with crucial guidance to navigate the wide range of processing choices. We recognise that TLS processing, and in particular voxel-based processing, is a fast-moving field. We also recognise, as the reviewer does, the strengths and weaknesses of different methods, and their different assumptions. No method is perfect, and we have made different choices to this reviewer, but that does not invalidate the work presented here. For example, whilst we agree that not all destructively validated methods will be applicable for all ecosystems, we do believe that processing methods that are destructively sampled present a reason to trust them over those that have not, and note that for many TLS developments, destructive sampling has been an important step towards acceptance of the technology (for example, for biomass estimates).

**1.1 By "broadly applicable", Flynn et al. mean "easy to apply to many kinds of datasets". However computationally easy to apply does not mean relevant. They choose a method which does not consider the scanning geometry to derive PAI from point density. Probably because they do not have that information at hand (?). This information is however in principle retrievable from the raw data (with some additional effort probably). The reason why this information is important seems to escape the authors. There is abundant literature which explains why it is crucial indeed to integrate scanning geometry in the analysis. I have previously shared some papers with the authors which make that point clear.**

We thank the reviewer for highlighting the importance of including scan geometry in the analysis. The importance of including scan geometry, as described by the reviewer, is to account for occlusion effects on PAI estimates. An important note on occlusion in the forest canopy is that they are "unknown unknowns". Even with sophisticated approaches that account for geometry, occlusion is still a problem because we do not know what we have not measured if it cannot be seen. For example, recent literature published (Zhu et al., 2023), highlight the assumptions made in radiative transfer models, for example, hard to measure correction factors and the assumption of a homogenous and turbid medium, arguing for a model that defines the 3D geometry of leaf material. Including the leaf 3D morphology would allow the representation of the structural properties of canopy vegetation to be considered that are difficult to include in a radiative transfer model. These assumptions may not hold

true for all vegetation types, leading to inaccuracies in PAI estimation. We also note that the approach we use, like those suggested by the reviewer, is essentially an implementation of Beer's Law, and we are therefore using assumptions in common to those proposed by the reviewer. From these conclusions, we draw attention to the active debate within the field, and a consensus to "best approach" for measuring PAI is far from being reached.

An important note on the papers suggested here by the reviewer, is sample size and computational limitations. The methods suggested by the reviewer have been developed with either a very small number of scanned trees, trees scanned with a scanning strategy so dense as to be impractical for most researchers (in this case to explore the effects of scan density and pattern on occlusion effects), or with simulated datasets. The dataset used in our study comprises 2472 trees scanned from 528 scan locations. While the effects of including scan geometry on PAI estimates is an important question, understanding the need for this added complexity, on a dataset of this size, is beyond the scope of this paper. We note that the dataset used in our study is freely available, and we would welcome any further exploration to this effect using our data.

**1.2 Essentially, the "point-based" method (which disregards the scanning geometry information) relies on very stringent assumptions of completeness of canopy sampling (no occlusion) and high quality of co-registration of multiple view scans. Conversely, methods based on radiative transfer are largely immune to slight displacement due to wind and imperfect co-registration effects and can accommodate some level of occlusion without bias (i.e. more robust, more widely applicable I would argue).**

We thank the reviewer for the comment on occlusion effects and co-registration error. We agree that these can both impact estimated PAI and have accounted for these in the following ways. The scanning strategy used in this study was on a 10 m grid, making it very dense for a Mediterranean ecosystem which has a relatively low canopy height and open canopy. By comparison, the 10 m grid strategy (Wilkes et al., 2017) was developed in a dense tropical forest and (Calders et al., 2018) used a 20 m grid in a UK closed canopy woodland. Our system is likely *more* open than the UK example, so we are confident that we have sampled conservatively to maximise information capture.

Occlusion effects are most prevalent when a single scan TLS strategy is applied in a structurally complex forest and is most effectively dealt with by using a multi scan approach (Wang and Fang, 2020) such as ours. We can therefore infer that occlusion effects in our dataset are low, especially when compared with the scanning strategies used in other TLS studies. When co-registering scans, we used a low threshold (0.015 m) to avoid error due to imperfect co-registration. Here we refer to (Owen et al., 2021) for more complete methodological detail.

**1.3 One of the arguments put forward to select the voxel method used (and to disregard alternative mainstream and, I believe, more sensible methods) is that the method selected has been validated with destructive sampling (which falsely suggests that other methods have not been tested against destructive sampling!). I am afraid that the authors did not ask themselves whether the method could have been validated in a particular context and might not be generally transposable. Actually, the study cited as reference which presents the validation protocol (Li et al 2016) only considered small (<3.2 m tall, < 9 cm dbh) magnolia trees (with very large leaves and thick twigs) scanned with no wind, at close range. In these very narrow validation conditions, the relative error in LAI was found to be about 20% and bias with size was clearly discernible. So the validation is at best weak.**

We thank the reviewer for highlighting the importance of destructive sampling and the uncertainty in ecological transferability, but we do not agree that we have suggested that this is the only method that has been validated by destructive sampling, and we do not think that this comment is provided in good faith. Whilst the data collected in Li et al. (2016) was not taken in exactly our ecosystem, the problem of transferability exists for any destructive sampling taken outside of the target study location and the accusation here could be levelled at a large number of high impact and valuable studies in this field. Large-scale destructive sampling spanning a breadth of ecotones has long eluded the remote sensing field, and almost all TLS papers apply methods that have not been tested with destructive sampling in precisely the same ecosystem. As we were unable to collect destructive samples in this study (which is common in ecological studies at all scales), and due to the known high sensitivity to voxel size of metrics derived via voxel methods, we picked a method that had a robust approach to determining voxel size while also including destructively sampled validation.

**1.4 In addition, and maybe more importantly, Li et al. explicitly state that the method they present is unlikely to apply to forest conditions where trees will be taller and occlusion will be much more important in which case the proposed method will be biased. These are precisely the conditions in the Flynn et al study. To make things worse it should be noted that Li et al use a long-range multiple return laser whereas the shorter-range single return Leica HDS6200 will penetrate less the vegetation thereby increasing the problem of occlusion.**

We thank the reviewer for highlighting known issues with occlusion in measuring PAI, and we of course considered whether the method of Li et al. (2016) could be applied to our study, so we regret the reviewer's misinterpretation of our work. The canopy height in our study region (Mediterranean woodlands) was well within the range of the Leica HDS6200 scanner used, and with the dense scanning strategy employed (see comment 1.2), occlusion in this dataset has been minimised. The Mediterranean ecosystem used here is not particularly dense, with many canopy gaps and low stem density. Furthermore, phase-shift systems, although nosier, have a higher scanning density for a given scan time than time-of-flight scanners which means that we are unlikely to see less vegetation penetration using this instrument. Minimising occlusion is a complex interplay between canopy structure, scan positions and equipment settings and vegetation penetration is maximised by both the beam frequency and number of scan positions. We were well aware of such potential issues during field data collection and took steps to minimise accordingly. Our dataset is openly available online for any researcher to explore this issue themselves.

**1.5 Li et al. also question the applicability of their method to coniferous trees given that to choose the optimal voxel size (equal to the mean point-to-point distance) they assume that the distance between neighbouring points is less that the distance between neighbouring leaves. The rationale for picking the mean point-to-point distance as the voxel resolution is that considering smaller voxels would generate many false empty voxels (negative bias) while taking voxels larger than the mean distance between contiguous leaves will increase the number of void spaces considered to be opaque and generate a positive bias. But the latter situation will happen in the case of needle foliage even with very small voxels so their criterion is not applicable in such a case. Down-sampling to a minimum distance between points of 5cm as done by Flynn et al. can only increase bias in the case of small leaves. Importantly this points to the fact that the optimal voxel size in the selected method depends on leaf size and arrangement and this is another reason why the method will not generalize well to multispecies stands.**

We thank the reviewer for their comment on the role of voxel size on PAI estimates in multispecies stands. We agree that the "correct" choice of voxel size is difficult to determine, and that the effects on PAI estimates can be large. To our knowledge, no voxel approaches (whether proposed by the reviewer or not) have proposed a definitive and independently verified method for voxel size choice. Exploration (currently unpublished) of methods proposed by the reviewer suggest that some implementations produce results that are highly sensitive to voxel size choice, and we are concerned that some voxel approaches may not be reliable for TLS processing, including some published approaches.

In the absence of a robust method to choose a voxel size for heterogeneous forests using within-voxel radiative transfer, we selected a method with clear protocol for determining voxel size choice. Previous work has shown significant variability across voxel size in PAI estimates, especially across different forest scenes (Wang and Fang, 2020) such as in this study. The method chosen in this study showed stability in the ecosystem within which it was tested when voxel size matched point to point minimum distance.

If voxel size was to be defined by the structural properties of the voxel (i.e., species, ecological context etc.) then each voxel (or at least each scan) would have to be individually parameterized which is impractical at scale. This also means that each voxel would require independently collected species and ecosystem information as a precondition to computing LAI, rendering many datasets unusable. The issue of voxel size spans all methods, including current within-voxel radiative transfer ones, and we would welcome studies addressing this specific challenge; clarification is sorely needed for TLS users.

**1.6 In addition, when scanning on a regular grid inside a plot the scanning geometry necessarily creates a large range of local point density as the point density will vary with distance from the laser (and notably from bottom to top of canopy). The point-to-point distance becomes highly variable (contrary to what is observed in Li et al's setting).**

We thank the reviewer for their comment on the reliance of our chosen method on the uniformity of point density in point clouds. In common with many researchers, we followed published best practice in our scanning strategy. As outlined in section 2.5 of our proposed manuscript, we downsampled our point clouds to 0.05 cm from dense raw scans to achieve uniformity of point density (see Owen et al. 2021 for more detailed description of the downsampling process), which is standard protocol in many described TLS processing pipelines (Burt et al., 2019; Wilkes et al., 2017). The use of a regular grid within a plot is a widely used strategy to collect TLS data while minimising occlusion effects and irregularity of point density. We also note the use of a height-dependent statistical filter which we implemented to retain uniformity of point density through the canopy (Owen et al. 2021; see also response to comment 1.8 below).

**1.7 More generally the lidar data processing is not thoroughly described and I can find many loose ends. For instance, Flynn et al. don't explain how they treat empty voxels inside trunks (not correcting for the occlusion there might be a reason why they find an increasing leaf-to-wood ratio with increasing tree size).**

We are a little confused by the comment regarding empty space inside trunks, as the method of Li et al. is developed for trees. We refer to the response to previous comments on risk of occlusion, and that our data are available for inspection. Nevertheless, we have added a discussion point on this potential explanation for the observed increase in leaf-to-wood ratio with tree size. LiDAR data processing is described within our paper, and in further detail in the cited manuscript Owen et al. (2021).

**1.8 Another example is the way they process noise points in their raw scans. The Leica HDS6200 is a single-return phase-based scanner. Phase-based scanners are typically faster than time of flight scanners but suffer from a high level of noise which has to be filtered. Doing so some true points are necessarily lost (which in many applications is not an issue given the very high point density such scanners can collect in a short time).**
**But when estimating PAI (whatever the method considered) an additional calibration step is needed to correct for the censorship bias introduced by the noise filtering. I could not find any information on how those noise points were dealt with. This should be presented for the sake of clarity and repeatability.**

We thank the reviewer for their comment regarding noise associated with phase-shift scanners. We agree that noise is a known problem particularly with phase-shift scanners but would not argue that this means that all data collected with phase-shift scanners are not valuable, and we have carefully followed standardised filtering and data processing approaches. All scanner technologies have known strengths and issues, and no sensor is perfect. Further, as outlined in section 2.4, we applied a height-dependent statistical filter to remove noise points. The significance of applying a height-dependent filter is that strength of the filter weakens with height. This is important where data has been collected from a ground-based instrument and there is more noise closer to the ground. A more comprehensive discussion on the role of height-dependent statistical filters can be found in Owen et al. (2021).

Although noise may be more pronounced in phase-shift systems, the problem persists regardless of instrument used. For example, Calders et al. (2018), outline the problem of partial hits always being classified as full hits using a time-of-flight system. This means that an overestimation in PAI is likely regardless of the method used. Additionally, Wilkes et al. (2021) scanned individual branches in laboratory conditions to estimate biomass, finding that even under ideal scanning conditions, bespoke filtering was still required to minimise the impact of partial beam hits. This is important as noise is a known problem regardless of instrument and environment and makes destructive or otherwise independent validation all the more important. In this study we present a robust approach to filtering of noise while limiting removal of vegetative points.

**1.9 In any case, the methodological flaws and uncertainties are too many to meaningfully discuss the ecological results. I am disappointed that the authors did not consider seriously my previous comments. I still hope they can improve on their analysis scheme because the data collected is indeed significant and might bring valuable ecological insights if rigorously processed.**

We seriously considered all points raised by the reviewer in the previous response and made several changes and clarifications. We regret that the reviewer does not recognise the significant effort we have made, and we are grateful for the other reviewers' and the editor's overwhelmingly positive assessment of our work.

**Referee 2**

**2.1 Please unify Beer-Lambert's law (Line 99) and Beer-Lambert law (Line 52).**

We thank the reviewer for identifying the inconsistency in our manuscript and have changed accordingly.

**2.2 Unifying the reference format through the article would be better. For example: Li et al., (2016) to (Li et al., 2016).**

We thank the reviewer for their suggestion of unifying the reference system throughout the manuscript. We have followed the guidelines published by Biogeosciences for in-text referencing and note the difference between these and references at the end of a sentence. We do note our incorrect use of comma in in-text references and have modified accordingly throughout the manuscript.

**2.3 Please unify the unit used in the manuscript, such as 5 cm and 0.05 m.**

We thank the reviewer for suggesting this edit and have unified the units used throughout the manuscript.

**2.4 L262 You would better change the variable as to φs, which makes readers distinguish it from α.**

We thank the reviewer for their suggestion of changing the variable in equation 2 and agree that φs is more distinguishable from α. We have updated this variable accordingly throughout the manuscript.

**2.5 Please note to add commas or full stops after equations.**

We thank the reviewer for drawing our attention to the lack of commas and full stops after equations and have added these after equations 1 and 2.

**2.6 Please recheck that the dashed line shown in Figure 3 is correct.**

We have checked the dashed lines and caption in Figure 3, noting that the dashed line in panel a represents 1:1 line and in panel b, 0. This is clearly stated in the figure caption. We have, however, amended the description of the regression line (solid black line) to better distinguish the SMA results from the dashed lines.

**2.7 Line 305 TLS-estimated. In Line 253, you defined CAI, so you can now use the abbreviation to refer to it more efficiently.**

We thank the reviewer for identifying the multiple definitions of CAI and have updates accordingly.

**2.8 Line 333 Please add a comma before "and PAI".**

We have added a comma before "and PAI".

**2. 9 Line347 Delete (CAI)**

(CAI) deleted.

**2.10 Line 364 Please reconsider and declare this sentence:**
**…trees in drier climates tend to have smaller leaves (Peppe et al., 2011), leading to more small canopy gaps…**

We thank the reviewer for pointing out the lack of clarity in this statement. We have amended the sentence to make clear our statement: "trees in drier climates tend to have smaller leaves, leading to more complex canopy gaps that TLS may resolve where DHP cannot."

**2.11 In my opinion, the voxel method is a more efficient way to extract local canopy structure features than searching for neighboring point clusters. However, this method may not capture all of the intricate details of the canopy structure.**

We thank the reviewer for their comment on the effectiveness of the voxel method we used. We completely agree that the voxel-based method may not capture all of the intricate details of the canopy structure, and this could be leading to an overestimation of PAI. We have added a statement to this effect in the discussion.

**2.12 Line 423 Change to "the negative relationships between height and α" and "the positive relationships between CAI and α".**

We thank the reviewer for their suggestion which makes our discussion point clearer. We have made the relevant changes to the manuscript.

**2.13 Please ensure that the number of authors listed in each reference is consistent.**

We thank the reviewer for their suggestion of keeping a consistent number of authors listed throughout the manuscript. We have followed guidelines published by Biogeosciences for single author, co-author and multi author papers and therefore the number of authors listed is consistent with the number of authors listed on each reference paper.

**2.14 Please label the values of b that appear in C2 to C4 in Appendix C. Also, remove the checkmarks that are used in these three equations.**

Checkmarks have been removed from equations and values of b have been added to table C4.

**2.15 In Tables B1, B2, C3 and C4, what are 95% CI and ICC?**

We have added "95% CI are 95% confidence intervals and ICC is the intra-class correlation coefficient" to table captions.

**Referee 3**

**3.1 L34-41 describes the TLS can estimate PAI, WAI, and LAI; whereas the intercomparison between different algorithms is lacking. Then, L42-82 focuses on DHP. However, there is no link between TLS and DHP. Maybe integrating the L34-41 into Section 1.2 is more suitable.**

We thank the reviewer for pointing out the confusing structure of the introduction and have moved the first paragraph of the introduction to section 1.1.

**3.2 L33 is "1 Introduction", however, L83 is "1.2 TLS methods for calculating PAI, LAI and WAI". Please check the chapter number.**

We thank the reviewer for identifying the numbering mistake in section 1. We have now updated the numbers in section 1.

**3.2 L146 41°23′N 4°21′W −> 41°23′N, 4°21′W**

We have added a comma to the coordinates described in section 2.1.

**3.3 L155 "33 30 x 30 m plots" is confused.**

We thank the reviewer for identifying the confusing language used to describe the plots. We have updated the manuscript to state: "33 plots of size 30 x 30 m"

**3.4 L272, space in p<0.001, please check all in the manuscript.**

We thank the review for identifying the lack of spaces in statements of p values. We have amended all instances in the manuscript accordingly.

**3.5 L276, slope = -0.88 −> slope = −0.88**

We thank the reviewer for identifying the incorrect symbol and have updated the hyphen to minus symbol.

**3.6 Fig. 3a and Fig. 4a, please use the same scale for x and y axis.**

We have now unified the axis scales in Figures 3a and 4a.

**References**

Burt, A., Disney, M., and Calders, K.: Extracting individual trees from lidar point clouds using treeseg, Methods Ecol. Evol., 10, 438–445, https://doi.org/10.1111/2041-210X.13121, 2019.

Calders, K., Origo, N., Disney, M., Nightingale, J., Woodgate, W., Armston, J., and Lewis, P.: Variability and bias in active and passive ground-based measurements of effective plant, wood and leaf area index, Agr. Forest Meterol., 252, 231–240, https://doi.org/10.1016/j.agrformet.2018.01.029, 2018.

Li, Y., Guo, Q., Tao, S., Zheng, G., Zhao, K., Xue, B., and Su, Y.: Derivation, Validation, and Sensitivity Analysis of Terrestrial Laser Scanning-Based Leaf Area Index, Can. J. Remote Sens., 42, 719–729, https://doi.org/10.1080/07038992.2016.1220829, 2016.

Owen, H. J. F., Flynn, W. R. M., and Lines, E. R.: Competitive drivers of inter-specific deviations of crown morphology from theoretical predictions measured with Terrestrial Laser Scanning, J. Ecol., 109, 2612–2628, https://doi.org/10.1111/1365-2745.13670, 2021.

Wang, Y. and Fang, H.: Estimation of LAI with the LiDAR Technology: A Review, Remote Sensing, 12, 3457, https://doi.org/10.3390/rs12203457, 2020.

Wilkes, P., Lau, A., Disney, M., Calders, K., Burt, A., Gonzalez de Tanago, J., Bartholomeus, H., Brede, B., and Herold, M.: Data acquisition considerations for Terrestrial Laser Scanning of forest plots, Remote Sens. Environ., 196, 140–153, https://doi.org/10.1016/j.rse.2017.04.030, 2017.

Wilkes, P., Shenkin, A., Disney, M., Malhi, Y., Bentley, L. P., and Vicari, M. B.: Terrestrial laser scanning to reconstruct branch architecture from harvested branches, Methods in Ecology and Evolution, 12, 2487–2500, https://doi.org/10.1111/2041-210X.13709, 2021.

Zhu, Y., Li, D., Fan, J., Zhang, H., Eichhorn, M. P., Wang, X., and Yun, T.: A reinterpretation of the gap fraction of tree crowns from the perspectives of computer graphics and porous media theory, Front Plant Sci, 14, 1109443, https://doi.org/10.3389/fpls.2023.1109443, 2023.